# Plk1, upregulated by HIF-2, mediates metastasis and drug resistance of clear cell renal cell carcinoma

Maeva Dufies [1,2✉], Annelies Verbiest[3,4], Lindsay S. Cooley[5], Papa Diogop Ndiaye[2,6], Xingkang He[7], Nicolas Nottet[8], Wilfried Souleyreau[5], Anais Hagege[2,6], Stephanie Torrino [9], Julien Parola[2,6,10], Sandy Giuliano[2,6], Delphine Borchiellini[10], Renaud Schiappa[10], Baharia Mograbi[6], Jessica Zucman-Rossi [11], Karim Bensalah[12], Alain Ravaud[13], Patrick Auberger[8], Andréas Bikfalvi [5], Emmanuel Chamorey[10], Nathalie Rioux-Leclercq[14], Nathalie M. Mazure[8], Benoit Beuselinck[3,4], Yihai Cao [7], Jean Christophe Bernhard[15], Damien Ambrosetti[16] & Gilles Pagès[1,2,6✉]

Polo-like kinase 1 (Plk1) expression is inversely correlated with survival advantages in many cancers. However, molecular mechanisms that underlie Plk1 expression are poorly understood. Here, we uncover a hypoxia-regulated mechanism of Plk1-mediated cancer metastasis and drug resistance. We demonstrated that a HIF-2-dependent regulatory pathway drives Plk1 expression in clear cell renal cell carcinoma (ccRCC). Mechanistically, HIF-2 transcriptionally targets the hypoxia response element of the Plk1 promoter. In ccRCC patients, high expression of Plk1 was correlated to poor disease-free survival and overall survival. Loss-of-function of Plk1 in vivo markedly attenuated ccRCC growth and metastasis. High Plk1 expression conferred a resistant phenotype of ccRCC to targeted therapeutics such as sunitinib, in vitro, in vivo, and in metastatic ccRCC patients. Importantly, high Plk1 expression was defined in a subpopulation of ccRCC patients that are refractory to current therapies. Hence, we propose a therapeutic paradigm for improving outcomes of ccRCC patients.

---

[1] Centre Scientifique de Monaco, Biomedical Department, 8 quai Antoine Premier, 98 000 Monaco, Monaco. [2] LIA ROPSE, Laboratoire International Associé Université Côte d'Azur - Centre Scientifique de Monaco, Nice, France. [3] Department of General Medical Oncology, University Hospitals Leuven, 3000 Leuven, Belgium. [4] Laboratory of Experimental Oncology, Department of Oncology, KU Leuven 3000 Leuven, Belgium. [5] University of Bordeaux, INSERM U1029, 33600 Pessac, France. [6] University Côte d'Azur, Institute for Research on Cancer and Aging of Nice (IRCAN), CNRS UMR 7284; INSERM U1081, Centre Antoine Lacassagne, 06189 Nice, France. [7] Department of Microbiology, Tumor and Cell Biology, Karolinska Institutet, SE-171 77, Stockholm, Sweden. [8] University Côte d'Azur, C3M, Inserm U1065, 06204 Nice, France. [9] University Côte d'Azur, Institut de Pharmacologie Cellulaire et Moléculaire, CNRS UMR7275, 06560 Valbonne, France. [10] Centre Antoine Lacassagne, 06189 Nice, France. [11] Inserm, UMR-1138, Génomique fonctionnelle des tumeurs solides, IUH, 75010 Paris, France. [12] Centre Hospitalier Universitaire (CHU) de Pontchaillou Rennes, service d'urologie, 35000 Rennes, France. [13] Centre Hospitalier Universitaire (CHU) de Bordeaux, service d'oncologie médicale, 33000 Bordeaux, France. [14] Department of Pathology, University Hospital, 35000 Rennes, France. [15] Centre Hospitalier Universitaire (CHU) de Bordeaux, service d'urologie, 33000 Bordeaux, France. [16] University Côte d'Azur, Centre Hospitalier Universitaire (CHU) de Nice, Hôpital Pasteur, Central laboratory of Pathology, 06000 Nice, France. ✉email: maeva.dufies@gmail.com; gilles.pages@unice.fr

The majority of clear cell renal cell carcinoma (ccRCC) patients carry genetic aberrations of the *von Hippel-Lindau* (VHL) gene leading to genetic stabilization of hypoxia-inducible factor (HIF) transcription factor. The HIF pathway drives tumor development and progression in the VHL-inactivated ccRCC. HIF transcriptionally targets over 100 genes[1], and the loss-of-function of VHL induces constitutive HIF-1α/2α expression that markedly upregulated their targeted genes, including vascular endothelial growth factor (VEGF) and erythropoietin (EPO). Consequently, ccRCC is a hypervascularized tumor that carries frequent mutations in chromosome 3p, which affects an array of chromatin-remodeling genes, including *Polybromo 1* (PBRM1), *SET Domain Containing 2* (SETD2), and *BRCA1 Associated Protein 1* (BAP1)[2,3]. Tyrosine kinase inhibitors (TKI) primarily targeting VEGF receptors (VEGFRs) such as sunitinib are the first-line therapy for treating metastatic ccRCC[4]. Immune checkpoint inhibitors targeting the cytotoxic T-lymphocyte antigen-4 (CTLA4) the programmed cell death 1 (PD1) or its ligand PDL1 have also been approved as the first-line therapy in some countries[5]. Sunitinib inhibits angiogenesis by blocking VEGFRs. Interestingly, it also directly inhibits ccRCC cell proliferation through non-VEGFR-mediated pathways. Nevertheless, clinical benefits are limited and transient in most cases and the majority of patients develop resistance over time[6,7].

Based on gene expression, methylation status, mutation profile, cytogenetic anomalies, and immune cell infiltration, 4 subtypes of ccRCC patients (ccrcc1–4) have been classified[8–11]. These markers have prognostic and predictive values for guiding TKI-based therapy. The ccrcc2&3 subtypes possess a good prognosis value of progression-free survival (PFS) and overall survival (OS), and are favorable for TKI therapy, whereas the ccrcc1&4 subtypes have the opposite prognostic values with poor prognosis and TKI responses. The ccrcc2-tumors often express proangiogenic genes, and ccrcc3-tumors resemble gene expression profiling of healthy kidney tissue. The ccrcc4-tumors exhibit an immune-inflamed phenotype, but an exhausted tumor cell capacity by immune cells. The ccrcc1-tumors belong to an immune-cold phenotype almost without lymphocyte infiltration[8–11]. Therefore, the ccrcc2&3-tumors are favorable for TKI therapy and the ccrcc4-tumors are potentially beneficial responders to immunotherapy. In contrast, ccrcc1-tumors fail to respond to either therapies[8,9]. Sunitinib-resistant tumor cells acquire an enhanced ability to proliferate. Therefore, cell-cycle regulators may be perturbed in sunitinib-resistant ccRCC tumors. Polo-like kinase 1 (Plk1) is a serine/threonine kinase that acts during cell cycle progression[12]. Plk1 inhibits p53, and p53 represses the Plk1 promoter[13]. High Plk1 expression correlates with advanced disease stage, histological grades, metastatic potentials, and short-term survival in various tumors[14,15]. The Plk1 inhibitor volasertib inhibits a variety of carcinoma cell lines and induces tumor regression in several experimental tumor models[16,17].

In this study, we describe a molecular mechanism of the HIF-2-Plk1-mediated ccRCC metastasis and drug resistance. Plk1 may also serve as prognostic marker to predict ccRCC progression and drug resistance. We propose a theranostic paradigm by targeting Plk1 for treating sunitinib-resistant ccRCC. We provide compelling experimental evidence to support our conclusions and relate our findings to clinical relevance.

## Results

**High levels of Plk1 mRNA correlate with the HIF pathway in various cancers.** Plk1 expression correlated with shorter PFS and OS in most of the cancers[18] (Supplementary Table 1). An in silico analysis revealed the presence of a consensus hypoxia response element (HRE) in the *Plk1* promoter (Supplementary Fig. 1a).

Scrutinization of the promoter (several kb upstream of the transcription start site and several kb in the 3′ end of the gene) did not reveal the presence of another consensus site for HIF binding except the one described in Supplementary Fig. 1 (ACGTG with a CACA repeat). Since HIF-1α and HIF-2α are regulated by protein stabilization, we investigated the correlation between Plk1 and mRNA levels of HIF-1/2 major targets representative of their activity rather than with HIF-1α or HIF-2α mRNA levels (Ca9 (HIF-1α), Oct4 (HIF-2α), or Glut1 (HIF-1α and HIF-2α)) in the TCGA PanCancer Atlas Studies (Supplementary Table 1). Plk1 expression correlated with HIF-1α target in 15 out of 26 available data on cancer types (enough data for robust statistical analysis in 26 out of 32 available cancer types) like melanoma, two types of kidney, head and neck, lung, and pancreatic cancers. Plk1 expression correlated with HIF-2 targets in ccRCC and in testicular adenocarcinoma, with HIF-1α and HIF-2α targets in 6 out of 26 cancer types like breast cancer, liver cancer, and sarcoma. Plk1 was independent of HIF-1α and HIF-2α in stomach cancer, uterine cancer, and uveal melanoma. Plk1 expression depends, at least in part, on HIF-1, HIF-1 and HIF-2, or HIF-2 in most cancers.

**Plk1 is a marker of poor prognosis in ccRCC.** Because of VHL inactivation, ccRCC represents a paradigm to assess the relationship between Plk1 and HIFs-α, and the impact of Plk1 on ccRCC aggressiveness. We analyzed the link between Plk1 levels and survival in different cohorts of ccRCC patients. In a French cohort (111 M0 ccRCC patients, Table 1), Plk1 mRNA levels were higher in ccRCC samples as compared to healthy kidney (p<0.0001, Fig. 1a). This result was confirmed on samples of the TCGA cohort (Supplementary Fig. 2b). In 43 tumors out of 111 of the French cohort, the two alleles of the *VHL* gene were either deleted, mutated, or the *VHL* promoter was methylated resulting in transcriptional inhibition. Tumors with inactivation of the two alleles and/or promoter methylation presented higher Plk1 mRNA levels as compared to tumors with normal or with only one inactivated *VHL* allele (p = 0.05, Fig. 1b). In the 111 samples of the French cohort, Plk1 mRNA levels did not correlate to the

---

**Table 1 The characteristics of the ccRCC M0 patients included in the study.**

|  | Total | Low Plk1 | High Plk1 | *p* value |
|---|---|---|---|---|
| Number | 111 | 83 | 28 | |
| pT | | | | 0.204 |
| 1 | 60 (54.1%) | 48 (57.8%) | 12 (42.9%) | |
| 2 | 11 (9.9%) | 9 (10.8%) | 2 (7.1%) | |
| ≥3 | 40 (36%) | 26 (31.3%) | 14 (50%) | |
| pN | | | | 0.662 |
| 0 | 103 (92.8%) | 77 (92.8%) | 26 (92.9%) | |
| ≥1 | 8 (7.2%) | 6 (7.2%) | 2 (7.1%) | |
| pM | | | | NA |
| 0 | 111 (100%) | 83 (100%) | 28 (100%) | |
| Fuhrman grade | | | | 0.14 |
| 1 | 1 (0.9%) | 1 (1.2%) | 0 (0%) | |
| 2 | 50 (45%) | 42 (50.6%) | 8 (28.6%) | |
| 3 | 41 (36.9%) | 30 (36.1%) | 11 (39.3%) | |
| 4 | 19 (17.1%) | 10 (12%) | 9 (32.1%) | |
| DFS (months) / Progression (%) | NR / 29.7% | NR / 24.1% | 39.1 / 46.4% | <0.001 |
| OS (months) / Death (%) | 71.8 / 24.3% | NR / 18.1% | 63 / 42.9% | <0.001 |

Patient characteristics and univariate analysis with the Fisher's or $\chi^2$ test. Statistical significance (*p* values) is indicated (see Fig. 1).

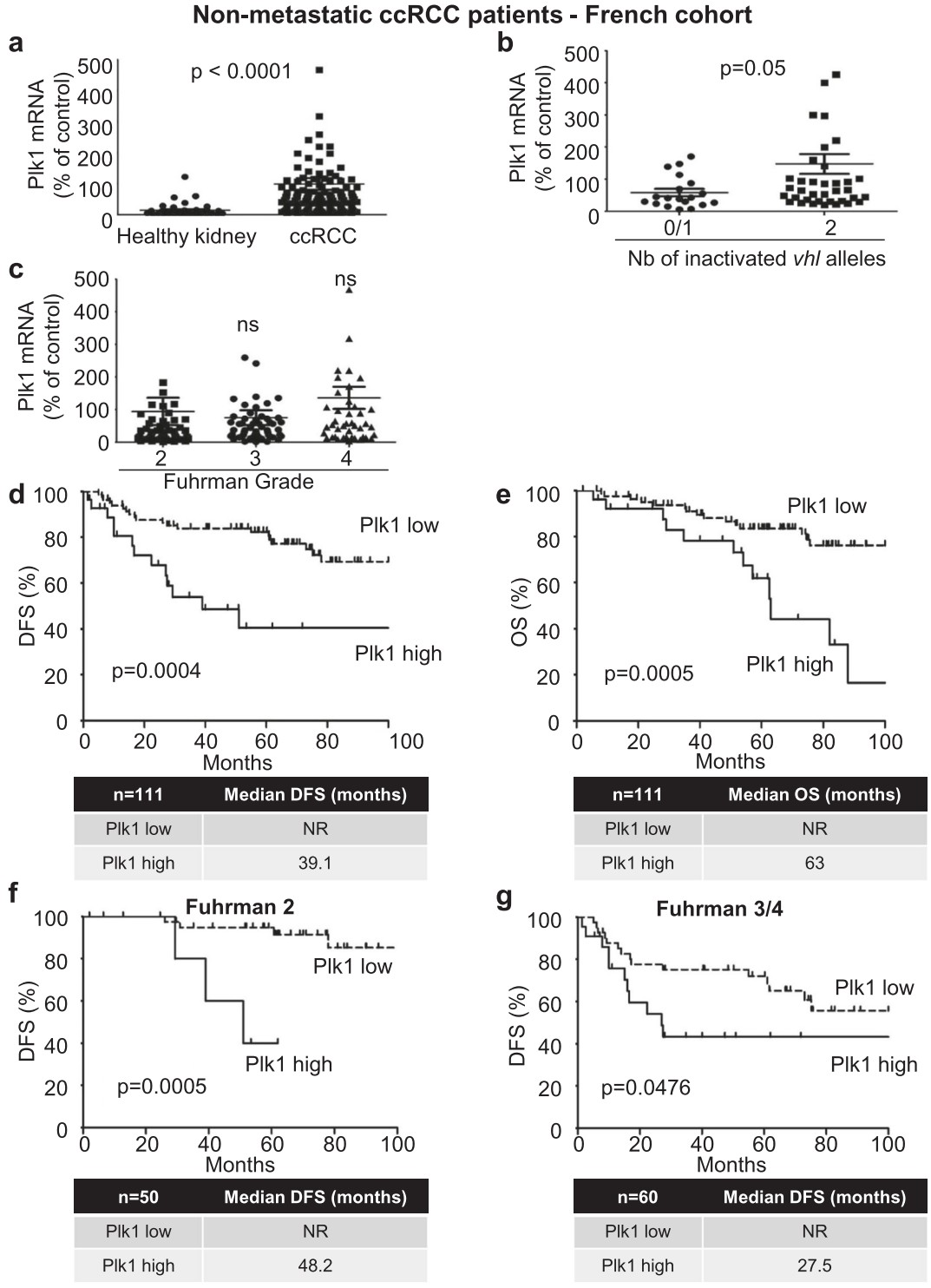

**Fig. 1 Plk1 is associated with poor prognosis in ccRCC.** 111 M0 ccRCC patients were analyzed for Plk1 mRNA levels in the kidneys (French cohort). **a** The levels of Plk1 mRNA in healthy kidney were compared with the levels in ccRCC. **b** The levels of Plk1 mRNA in ccRCC patients with VHL-WT (0 or 1 inactivated vhl allele) were compared to the levels in ccRCC patients with VHL-inactivated (2 inactivated vhl alleles). Vhl allele inactivation corresponds to a deletion, a mutation, or to a methylation of the promoter. **c** The levels of Plk1 mRNA were analyzed in different Fuhrman grade group (2–4). **d**, **e** The levels of Plk1 mRNA in 111 non-metastatic ccRCC patients correlated with DFS (**d**) or with OS (**e**). **f**, **g** The levels of Plk1 mRNA in non-metastatic low-grade (Fuhrman 2, **f**) or high-grade (Fuhrman 3 and 4, **g**) ccRCC patients correlated with DFS. The third quartile value of Plk1 expression was chosen as a cut-off. For (**a–c**), statistics were determined using an unpaired Student's *t* test. For (**d–g**), the Kaplan-Meier method was used to produce survival curves and analyses of censored data were performed using Cox models. Statistical significance (*p* values) is indicated (see Table 1).

**Table 2 Multivariate analysis — ccRCC M0 patients.**

|  | Description | Coef | z | p value |
|---|---|---|---|---|
| DFS |  |  |  |  |
| Biological parameter : Plk1 | Low vs High | 0.968 | 2.582 | 0.01 |
| Clinical parameters : Fuhrman grade | 1 and 2 vs 3 and 4 | 1.233 | 2.834 | 0.005 |
| OS |  |  |  |  |
| Biological parameter : Plk1 | Low vs High | 1.051 | 2.604 | 0.009 |
| Clinical parameters: Fuhrman grade | 1 and 2 vs 3 and 4 | 0.991 | 2.083 | 0.037 |

Multivariate analysis of Plk1, the Fuhrman grade, and PFS or OS. The multivariate analysis was performed using Cox regression adjusted to the Fuhrman grade.

Fuhrman grade (a score used by pathologists to define the aggressiveness of the tumor, Fig. 1c), but high levels of Plk1 mRNA correlated with shorter DFS (39.1 months vs >100 months, $p = 0.0004$, Fig. 1d) and OS (63 months vs >100 months, $p = 0.0005$, Fig. 1e). The correlation between high Plk1 mRNA levels and shorter OS of 376 M0 ccRCC patients was confirmed in samples of the TCGA cohort (74 months vs >150 months, $p < 0.0001$, Supplementary Fig. 2b). In the whole French cohort, Plk1 mRNA levels were also indicative of DFS for low-grade tumors (Fuhrman 2; 48.2 months vs >100 months, $p = 0.0005$, Fig. 1f) and high-grade tumors (Fuhrman 3 and 4; 27.5 months vs >100 months, $p = 0.0476$, Fig. 1g). Hence, in this cohort, Plk1 mRNA levels represented an independent marker for DFS and OS, of the Fuhrman grade in multivariate analyses (Table 2).

mRNA and protein levels are not always correlated. To show that Plk1 protein levels, as Plk1 mRNA levels, correlated with survival in ccRCC, Plk1 protein levels were analyzed by IHC on tissue microarrays (TMA cohort of 131 samples (101, M0 and 30, M1) of ccRCC patients, Supplementary Table 2 and Supplementary Fig. 3a). Plk1 expression did not correlate to the Fuhrman grade (Supplementary Fig. 3b, same results obtained on the analysis of Plk1 mRNA level, Fig. 1c) or with the metastatic stage (Supplementary Fig. 3c). These results confirmed those obtained on mRNA: high Plk1 protein levels correlated with a shorter DFS ($p = 0.042$) and OS ($p = 0.0243$) in M0 patients (Supplementary Fig. 3d, e) and with a shorter PFS ($p = 0.0492$) and OS ($p = 0.0272$) in M1 patients (Supplementary Fig. 3f, g). The correlation between high Plk1 mRNA level and shorter OS of 70 M1 ccRCC patients was confirmed on samples of the TCGA cohort (Supplementary Fig. 2c). Hence, by two independent approaches based on the evaluation of Plk1 mRNA or protein levels and on several cohorts of patients, we showed that Plk1 is a robust biological prognostic marker of survival in M0 and M1 patients, independent of clinical parameters including the metastatic status and the Fuhrman grade.

**HIF-2α binds to the Plk1 promoter and stimulates its activity in RCC cells.** The relationship between hypoxia and Plk1 expression was further assessed in human RCC kidney tumor cell lines (RCC4 (R4), RCC10 (R10), 786-O (786), A498 (498), ACHN (A), Caki2 (C2); Fig. 2a) and human primary normal (15S) and human primary confirmed RCC cells (TF, MM, CC; Fig. 2b)[19]. HIF-1/2α expression was null or very low in cells with active VHL (A and C2, TF and 15S). Cells inactivated for VHL (VHL-i) expressed HIF-1α and HIF-2α (R4, CC), or only HIF-2α (R10, 786, 498, MM). Cell lines and primary normal or tumor cells with an active VHL (A, C2, 15S, TF) have undetectable or low levels of Plk1 as compared to VHL-i cell lines and primary cells (R4, R10, 786, 498, MM, CC; Fig. 2a, b). Chromatin immunoprecipitation (ChIP) showed that HIF-2α bound to the domain of the Plk1 promoter containing the unique consensus binding site for HIF binding in two independent cell lines (786 (left) and R4 (right); Fig. 2c and Supplementary Fig. 1). However, in R4 cells expressing both HIF-1α and HIF-2α, ChIP experiments failed to show any HIF-1αβ binding on the above-mentioned domain of the Plk1 promoter (Fig. 2c, right). These results suggest a direct regulation of Plk1 transcription by HIF-2 but not by HIF-1 in ccRCC cells. Our results are consistent with those of publicly available ChIP-Seq data sets showing that no HIF1 binding was detected on the promoter and the 3' end of the Plk1 gene[20].

The role of HIF-1α and HIF-2α in Plk1 transcription was evaluated by testing Plk1 promoter activity using the luciferase reporter assays. It was further assessed by evaluating Plk1 mRNA levels after HIF-1α and/or HIF-2α downregulation in VHL-i cells or after hypoxia in VHL-WT cells.

HIF-2α-directed siRNA (siH2) decreased Plk1 promoter activity (Fig. 2d and Supplementary Fig. 4a) and Plk1 mRNA levels (Fig. 2e and Supplementary Fig. 4b) in VHL-i cells expressing only HIF-2α (786, R10, 498). Equivalent results (decreased promoter activity and mRNA levels) were obtained for VHL-i primary ccRCC cells expressing only HIF-2α (MM, Supplementary Fig. 4c, d).

In primary RCC cells expressing both HIF-1α and HIF-2α (CC), HIF-2α downregulation significantly decreased the Plk1 promoter activity (Supplementary Fig. 4e) and the amount of Plk1 mRNA (Supplementary Fig. 4f). In these cells, siH1 had no significant effect on promoter activity and Plk1 mRNA levels, and no additive inhibition was obtained by combining siH1 and siH2 (CC, Supplementary Fig. 4e, f). Following HIF-α stabilization by hypoxia in RCC cell lines (A) or in primary RCC cells (TF) expressing active VHL, the Plk1 promoter activity and Plk1 mRNA levels were upregulated (Supplementary Fig. 4g–j).

These results in cell lines and primary cells strongly favored the notion that HIF-2, but not HIF-1, is a direct transcriptional regulator of Plk1.

To further demonstrate a direct correlation between the transcriptional regulation of the Plk1 promoter and expression of the Plk1 protein, by immunoblotting we tested Plk1 protein expression together with HIF-1α and HIF-2αβ in siH2- and VHL-transfected cells.

siH2 inhibited Plk1 protein level in VHL-i RCC cell lines (R10, 498, 786, Fig. 2f) and in MM primary RCC cells expressing only HIF-2α (Fig. 2G). Re-introduction of a functional VHL in 786 cells (VHL) decreased Plk1 levels (Fig. 2f).

Hypoxia stabilized HIF-2α in A (Fig. 2h) and in TF primary cells (Fig. 2i) resulting in Plk1 protein induction. In A cells, hypoxia-induced over-expression of Plk1 was reverted by siRNA against HIF-2 (Fig. 2h). Hence, hypoxia or VHL inactivation drives Plk1 protein expression via a transcriptional program involving a HIF-2α-dependent stimulation of its promoter.

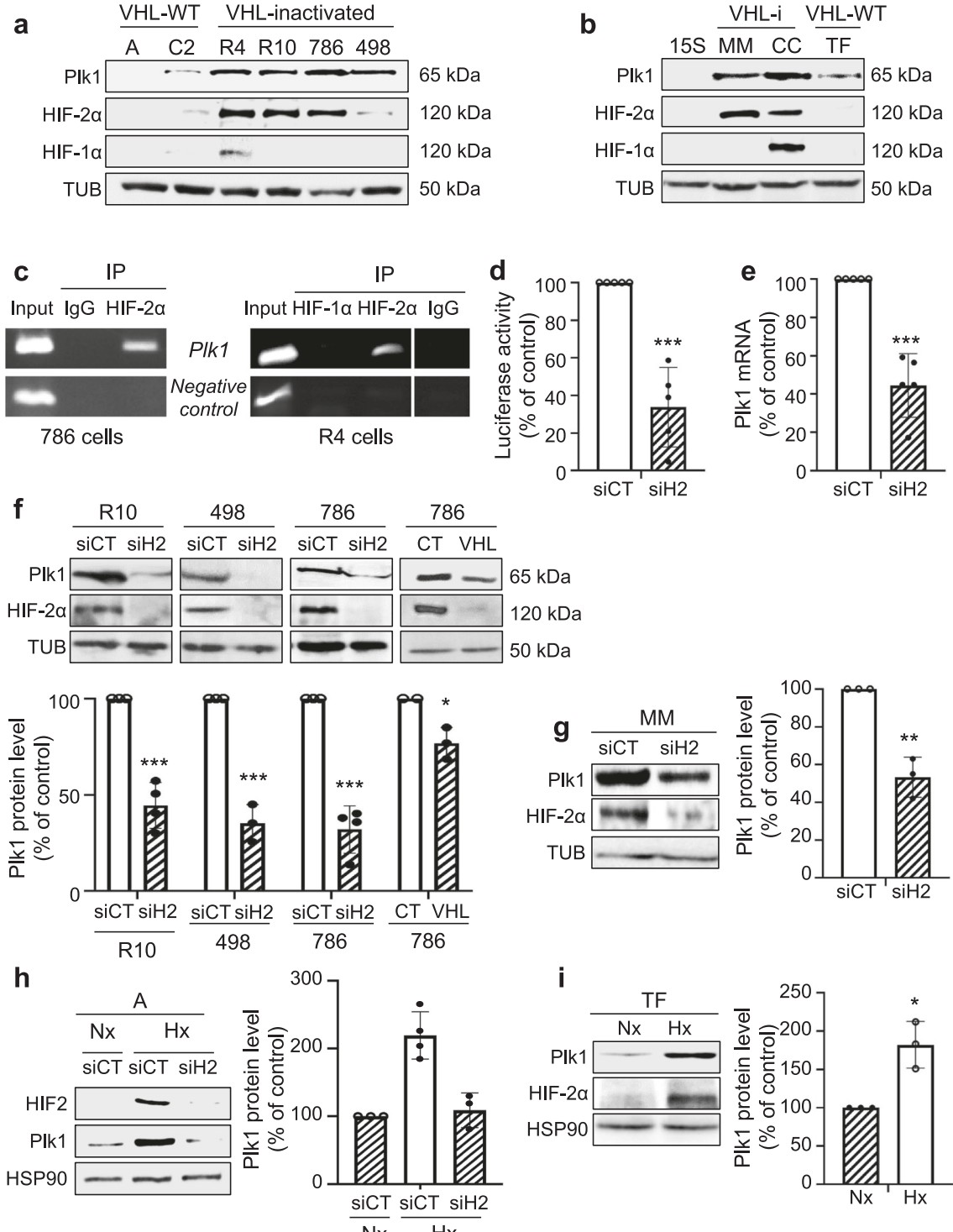

**SETD2 mutation correlates with Plk1 increased expression in ccRCC cells inactivated for VHL.** In all, 80% of ccRCC are inactivated for VHL but chromatin-remodeling genes (PBRM1, BAP1, and SETD2) are also frequently mutated. Mutations in PBRM1 and/or BAP1 did not modify Plk1 expression. However, tumors of the TCGA database inactivated for VHL and SETD2, presented higher mRNA levels as compared to tumors inactivated only for VHL (Fig. 3a). Tumors with the wild-type (WT) or the mutated forms of SETD2 expressed equivalent Plk1 mRNA levels in a VHL-WT context (Fig. 3b). We therefore examined the

mutational status of SETD2 in our RCC cell lines with inactivated VHL or VHL-WT. The 786, A, and C2 cells express normal SETD2, and 498 cells present an inactivating mutation that does not affect its expression (*SETD2* V2536Efs*9; TCGA and CCLE data). SETD2 downregulation by siRNA in 786 cells increased the mRNA and protein levels of Plk1 while it had no effects in 498 cells (Fig. 3c, e). SETD2 downregulation in VHL-WT cells did not alter Plk1 mRNA and protein levels (Fig. 3d, f). Our results suggested that the presence of an active SETD2 controls Plk1 expression. Inactivation of SETD2 further stimulates Plk1

**Fig. 2 HIF-2 bond to the Plk1 promoter and regulated its expression in ccRCC cells. a, b** Different RCC cell lines [(ACHN (A), Caki2 (C2), RCC4 (R4), RCC10 (R10), 786-O (786), and A498 (498)] (**a**) or primary RCC cells (TF, MM, and CC) and healthy renal cells (15S) (**b**) were evaluated for Plk1, HIF-1α, and HIF-2α expression by immunoblotting. Tubulin (Tub) served as a loading control. **c** ChIP experiments with HIF-2α and HIF-1α antibodies or negative CT antibodies were performed on extracts from 786 (right) and R4 (left) ccRCC cells. The promoter region of the *Plk1* promoter containing the HIF-α binding site was amplified by PCR. Results are representative of three independent experiments. **d** ccRCC cell lines (VHL-i) 786 were transfected with siRNA against HIF-2α (siH2) for 24 h. Cells were then transfected with a *Renilla luciferase* reporter gene under the control of the Plk1 promoter. The *R. luciferase* activity normalized to the firefly luciferase (control vector) was the read-out of the Plk1 promoter activity. **e** 786 cells were transfected with siRNA against HIF-2α (siH2) for 48 h. The Plk1 mRNA level was determined by qPCR. **f, g** VHL-i RCC cell lines (R10, 498, 786, **f**), or primary RCC cells (MM, **g**) were transfected with siRNA against HIF-2α (siH2) for 48 h. Plk1 and HIF-2α expression was evaluated by immunoblotting. HSP90 served as a loading control. The graphs show the level of Plk1. Control conditions were considered as the reference value (100). **h** VHL-WT RCC cell lines (A) were transfected with siRNA against HIF-2α (siH2) for 24 h and then were cultured in normoxia (Nx) or hypoxia 1% $O_2$ (Hx) for 24 h. Plk1 and HIF-2α expression was evaluated by immunoblotting. HSP90 served as a loading control. The graphs show the level of Plk1 (mean of three experiments). Control conditions were considered as the reference value (1). **i** VHL-WT primary RCC cells (TF) were cultured in normoxia (Nx) or hypoxia 1% $O_2$ (Hx) for 24 h. Plk1 and HIF-2α expression was evaluated by immunoblotting. HSP90 served as a loading control. The graphs show the level of Plk1 (mean of three experiments). Control conditions were considered as the reference value (1). Results are the means of three or more independent experiments (biological replication) represented as mean ± SEM. Statistics were determined using an unpaired Student's *t* test: $^*p < 0.05$, $^{**}p < 0.01$, $^{***}p < 0.0001$.

expression in cells presenting a constitutive expression of HIF-2α. The molecular link between SETD2 and HIF-2α was further examined. siH2 or VHL-WT increased SETD2 mRNA and protein levels in 786 cells (VHL-i, constitutive HIF-2α; Fig. 3g, i). On the contrary, hypoxia decreased SETD2 mRNA and protein levels in A and C2 VHL-WT cells in which hypoxia induces HIF (Fig. 3h, j). These results suggested that in hypoxia or following complete VHL inactivation, SETD2 downregulation leads to a HIF-2-dependent stimulation of *Plk1* expression. Hence, an enhanced aggressiveness program involves Plk1 upregulation through SETD2 inactivation and HIF-2α stabilization.

**Plk1 promotes an invasive phenotype and induces sunitinib resistance.** The link between Plk1 and ccRCC aggressiveness was assessed through the analysis of the TCGA database. A volcano plot showed 933 upregulated (4.3%) and 316 downregulated (1.5%) genes in tumors expressing high or low levels of Plk1 (Supplementary Fig. 5a). Hierarchical cluster analyses showed distinguishable expression profiles for tumors expressing high (red color) or low (blue color) levels of Plk1 (Supplementary Fig. 5b). High levels of Plk1 positively correlated with high proliferation, strong invasive potential, and resistance to the p53-dependent cell death pathway (Supplementary Fig. 5c, d).

The role of Plk1 in the aggressiveness of ccRCC was assessed by two strategies: inactivation/downregulation or over-expression of *Plk1*. Any attempt to stably downregulate or to invalidate *Plk1* failed. Hence, we stably transfected 786 cells with a *Plk1* expression vector. Two independent clones expressing 2.5-fold more Plk1 than the basal levels of empty vector (EV)-transfected cells were obtained (786 *EV*; 786 *Plk1-1* and 786 *Plk1-2*) (Fig. 4a). Characteristic parameters of aggressiveness were tested in these cells. While increased expression of *Plk1* did not result in enhanced proliferation rates (maximal potential reached by control cells), it stimulated cell migration abilities suggesting an increased ability to metastasize (Fig. 4b). Sunitinib-treated naïve ccRCC cells exhibited characteristics of inhibition of cell proliferation, G1-S cell cycle arrest, and DNA damage response attributed to p53 activation and senescence[21]. The viability and death of cells with increased levels of Plk1 were affected to a lesser extent by sunitinib (Fig. 4c, d). Sunitinib activated p53 (total and phosphorylated form (p-p53), Fig. 4e) and senescence (Fig. 4f) in 786 *EV* but not in 786 *Plk1-1* and 786 *Plk1-2* cells. Plk1 expression was increased in tumor samples of metastatic patients treated with sunitinib in a neoadjuvant setting (a rare procedure for inoperable patients bearing large tumors, 8 samples out of 3000 samples of ccRCC from 10 cancer centers in France[22]) and the relapse rapidly followed surgery although sunitinib, the

reference treatment, was maintained as an adjuvant treatment[22] (Supplementary Fig. 6a).

The relationship between Plk1 expression and sunitinib resistance was further addressed in sunitinib-resistant cells (786 R). Plk1 expression was higher in 786R as compared to naïve 786 cells (Fig. 4g). We previously showed the importance of the p38 MAP Kinase (p38) activity in resistance to sunitinib[22]. Therefore, we hypothesized that p38 was involved in Plk1 expression. A p38 inhibitor decreased Plk1 mRNA and protein levels in 786R cells (Supplementary Fig. 6b, c). These results strongly support the notion that Plk1 is a key player in the mechanisms of resistance to sunitinib by bypassing senescence and by promoting dissemination capabilities.

**Volasertib induced the death of resistant ccRCC cells and of primary ccRCC cells.** Since we showed that Plk1 is a key in ccRCC aggressiveness, we examined the sensitivity of ccRCC cells to volasertib. ccRCC cell lines and primary ccRCC cells were more sensitive to volasertib as compared to normal kidney cells (Supplementary Table 3). At low concentrations, volasertib decreased the proliferation and induced the death of naïve and sunitinib-resistant ccRCC cells (R10, R10R, 786, 786R, and 498; Supplementary Fig. 6d, e). Same results were obtained by using a specific siRNA against Plk1. Volasertib-mediated cell death was mainly due to mitotic catastrophe (MC; increased polyploidy, Supplementary Fig. 6f). Effects of volasertib on different parameters in 786, 786R, R10, and R10R cells were assessed . It decreased their viability (Fig. 4h) and their clonogenic potential (Supplementary Fig. 6d). It induced apoptosis (Fig. 4i) through caspase 3 activation (Fig. 4j) and increased abnormal mitosis and cytokinesis (hematoxylin-eosin (HE) staining, Fig. 4k).

Volasertib decreased the clonogenic potential (Supplementary Fig. 7a), the viability (Supplementary Fig. 7b), induced cell death (Supplementary Fig. 7c), and caspase 2 activation (Supplementary Fig. 7d) of primary ccRCC cells (CC, MM, TF) while it had no effects on normal 15S cells (Supplementary Fig. 7b–d)

These results suggest that Plk1 is intrinsically active on ccRCC cell lines and primary cells, and also reverts sunitinib resistance.

**Volasertib has a strong anti-tumor effect in experimental ccRCC in mice, in a model of metastasis in the zebrafish, and on primary tumor fragments.** Following in vitro characterization, effects of volasertib in integrated experimental models were assessed. Volasertib inhibited the growth of experimental tumors in mice more efficiently than sunitinib (Fig. 5a). Control tumors (CT) were heavier as compared to tumors from volasertib-treated

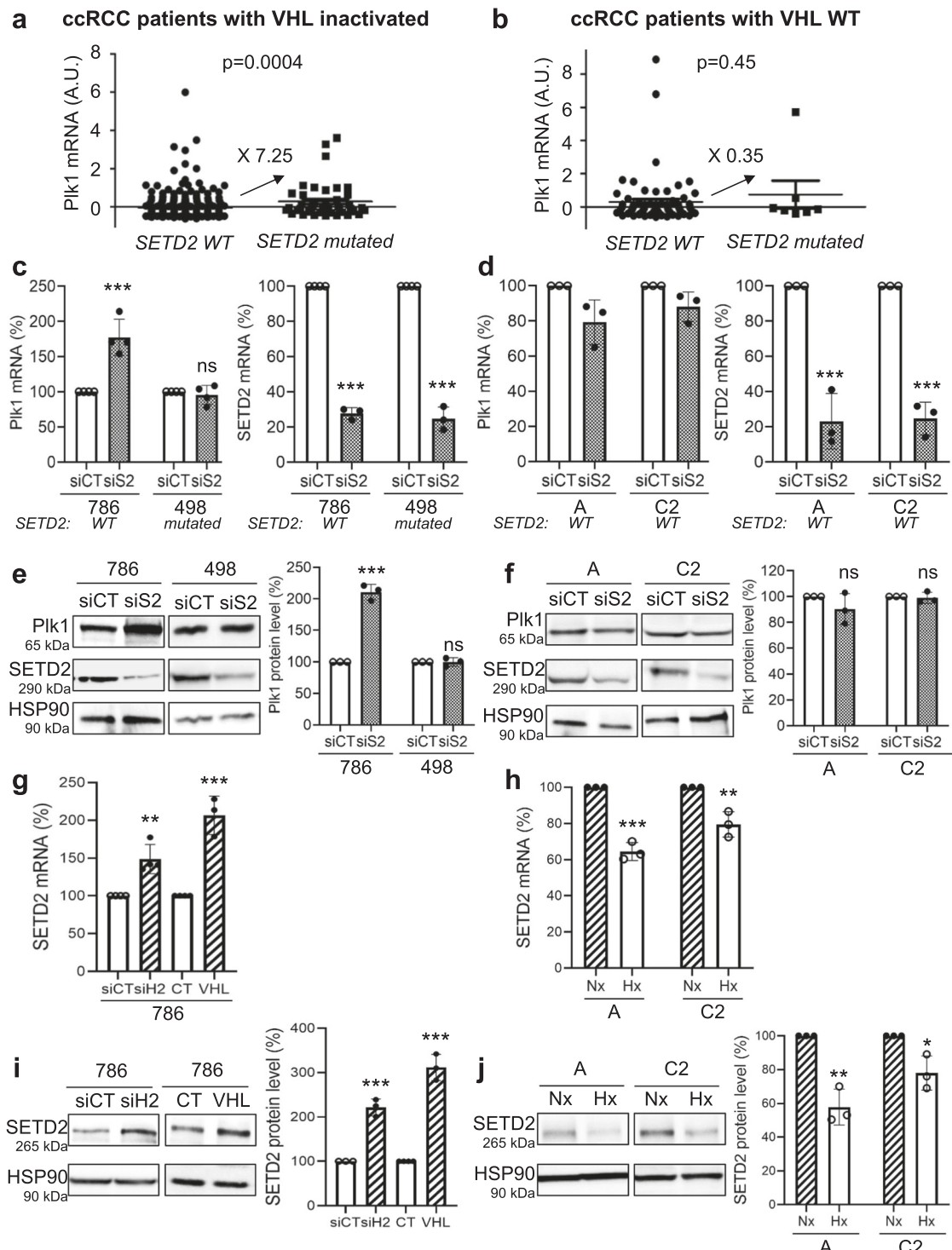

**Fig. 3 SETD2 inactivation induced Plk1 expression in ccRCC VHL-i cells. a, b** Levels of Plk1 mRNA (z-score) in ccRCC patients with wild-type (WT) SETD2 were compared to the levels in ccRCC patients with mutated SETD2, in RCC patients with VHL-i (**a**), or in ccRCC patients with VHL-WT (**b**). **c–f** 786, A, and C2 have WT SETD2 gene while 498 have a mutated/inactivated SETD2 gene (TCGA and CCLE data). VHL-i 786 and 498 RCC cells (**c**, **e**), or VHL-WT RCC cells A and C2 (**d**, **f**) were transfected with siRNA against SETD2 (siS2) for 72 h. The Plk1 and SETD2 mRNA levels were determined by qPCR (**c**, **d**). Plk1 and SETD2 expression was evaluated by immunoblotting. HSP90 served as a loading control (**e**, **f**). **g**, **h** 786 cells (VHL-i) were transfected with siH2 for 48 h or an expression vector coding for VHL (stable expression, 786+VHL). SETD2 mRNA levels were determined by qPCR (**g**). SETD2 expression was evaluated by immunoblotting. HSP90 served as a loading control. The quantification of Plk1 expression (mean of three experiments) is shown (**h**). **i**, **j** VHL-WT A and C2 cells were cultured in normoxia (Nx) or hypoxia 1% $O_2$ (Hx) for 24 h. SETD2 mRNA levels were determined by qPCR (**i**). SETD2 expression was evaluated by immunoblotting. HSP90 served as a loading control. The quantification of Plk1 expression (mean of three experiments) is shown (**j**). The value of the control condition was considered as the reference value (100). Results are the means of three or more independent experiments (biological replication) represented as mean ± SEM. Statistics were analyzed using an unpaired Student's t test: $^*p < 0.05$, $^{**}p < 0.01$, $^{***}p < 0.001$.

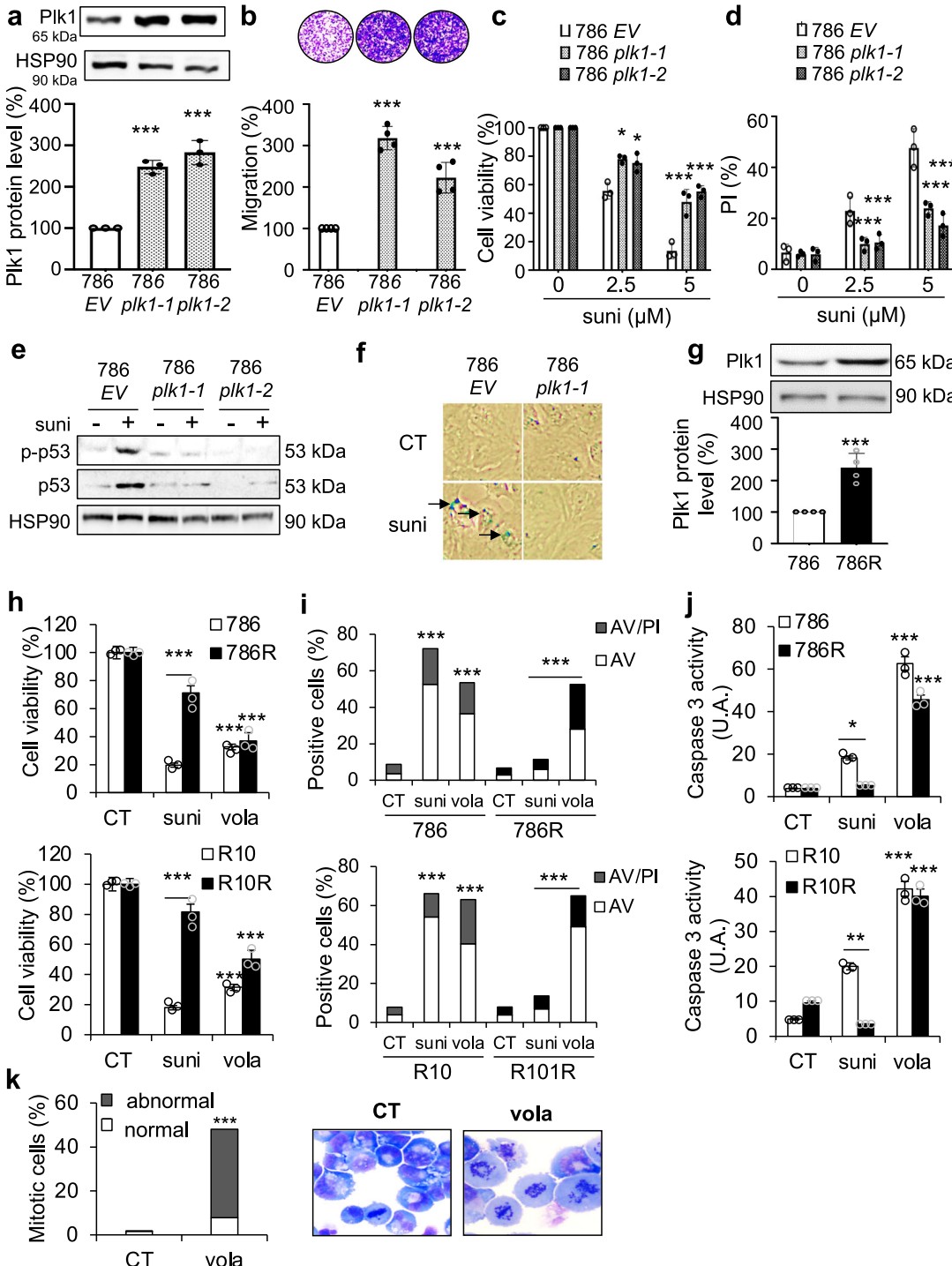

mice (Fig. 5b), in which a decreased number of proliferative cells (Ki67 staining, Fig. 5c) and mitotic defects (HES, Supplementary Fig. 8a) were observed. Volasertib also decreased the number of blood vessels reaching the tumors, and their density (Fig. 5d, e; decreased mRNA levels of endothelial cells (CD31) and pericytes (αSMA, marker of vessel integrity); Supplementary Fig. 8b). These results suggest that volasertib, in addition to its intrinsic effects on tumor cells, inhibits proliferative endothelial cells and pericytes. Hence, in this context, it can be considered as an inhibitor of angiogenesis.

The zebrafish was used as a relevant model of metastasis by assessing dissemination of tumor cells from the site of injection to the tail[23]. In this model, 786R had a strong ability to metastasize in the tails of zebrafishes. While sunitinib was unable to inhibit the formation of metastases in the tails of the zebrafishes, volasertib significantly reduced their size and number (Fig. 5f, g).

For a theranostic approach that can be implemented in the clinic, volasertib efficacy was tested on sections of tumors obtained from surgical specimens (Fig. 5h). Sunitinib and volasertib decreased the viability of tumor fragments in a dose-dependent manner although the effects of volasertib were more important for the highest concentration as compared to those of sunitinib (Fig. 5i). However, HE staining showed that volasertib only induced necrosis (Fig. 5j and Supplementary Fig. 8c). These

**Fig. 4 Over-expression of Plk1 induced aggressiveness, resistance to sunitinib, and its inhibition by volasertib induced cell death. a** 786 cells were transfected with the empty vector (EV) or Plk1 expression vector, and two clones (786 Plk1-1 and 786 Plk1-2) stably expressing Plk1 were selected. Plk1 expression was evaluated by immunoblotting. HSP90 served as a loading control. The quantification of Plk1 expression (mean of three experiments) is shown. The value of the conditions with 786 cells were considered as the reference value (100). **b** Serum-stimulated cell migration was analyzed using Boyden chamber assays on 786 EV, 786 Plk1-1, and 786 Plk1-2 cells. The level of migration of 786 cells was considered as the reference value (100%). Representative images of the lower surface of the membranes are shown. **c**, **d** 786 EV, 786 Plk1-1, and 786 Plk1-2 cells were treated with 2.5 or 5 µM sunitinib (suni) for 48 h. Cell viability was measured with the XTT assay (**c**). Cell death was evaluated by flow cytometry. Cells were stained with PI. Histograms show PI-positive cells (**d**). **e** 786 EV, 786 Plk1-1, and 786 Plk1-2 cells were treated with 2.5 µM suni for 48 h. p-p53 and p53 expression was evaluated by immunoblotting. HSP90 served as a loading control. These results are representative of three independent experiments. **f** 786 EV and 786 Plk1-1 cells were treated with 2.5 µM suni for 48 h. Senescence were evaluated by β-galactosidase staining. **g** Plk1 expression was evaluated by immunoblotting in 786 and 786 cells resistant to sunitinib (786R). HSP90 served as a loading control. The quantification of Plk1 expression (mean of three experiments ± SEM) is shown. Plk1 expression in 786 cells served as the reference value (100%). **h–j** 786 and 786R or R10 and R10R cells were treated with 100 nM volasertib (vola) or 5 µM sunitinib (suni) for 48 h. Cell viability was measured with XTT assays (**h**). Cell death was evaluated by flow cytometry. Cells were stained with PI and AV. Histograms show AV$^+$/PI$^-$ cells (apoptosis) and AV$^+$/PI$^+$ cells (post-apoptosis or another cell death) (**i**). Caspase-3 activity was evaluated using Ac-DEVD-AMC as a substrate (**j**). **k** 786 cells were treated with 100 nM vola for 24 h. Hematoxylin-eosin (HE) staining was done assessment, and the number of cells with normal and abnormal mitosis was evaluated. Results are the means of three or more independent experiments (biological replication) represented as mean ± SEM. Statistics were determined using an unpaired Student's t test: $^*p < 0.05$, $^{***}p < 0.001$.

results support the relevance of volasertib as a therapeutic alternative to sunitinib for ccRCC.

**Plk1 expression and molecular ccRCC subtypes as indicators for therapy decision.** Analysis of the TCGA database and the Cancer Immunome Atlas (TCIA) showed that M1 ccRCC patients with a low expression level of Plk1 and PDL1 had the longest OS, patients with a high expression level of PDL1 had an intermediate OS, and patients with high Plk1 and low PDL1 had the shortest OS (Supplementary Fig. 9a). Although PDL1 is not a validated marker for therapy selection, an elevated PDL1 expression has been associated with a good response to immunotherapy in patients with metastatic ccRCC[5]. The Immuno-PhenoScore (IPS) determined by The Cancer Immunome Atlas (TCIA) supports these clinical observations. Over-expression of Plk1 and a low expression of PDL1 were associated with a poor IPS reflecting a reduced efficacy of immunotherapy (Supplementary Fig. 9b). The impact of Plk1 expression on the PFS of 158 primary ccRCC patients treated in the first-line treatment with TKI (sunitinib, pazopanib, and sorafenib; Table 3) was investigated. Plk1 expression was more important in tumors of patients with an intermediate and poor International Metastatic RCC Database Consortium Score (IMDC score) reflecting tumor aggressiveness according to clinical and biological parameters (Fig. 6a). Patients with an intermediate and poor IMDC score are poor responders to TKI[24]. Over-expression of Plk1 (superior to the third quartile) was correlated with a shorter PFS for patients treated in the first line with TKI (PFS of 4 months vs 11 months, $p = 0.0015$, for any TKI, Fig. 6b; PFS of 2 months vs 14 months, $p = 0.0008$, considering sunitinib treatment only, Fig. 6c). Patients with a poor IMDC score, a low expression of Plk1 were correlated to a longer PFS (PFS of 5 months vs 2 months, $p = 0.0076$, Fig. 6d). Patients treated in the first line with TKI also have a short OS when Plk1 is over-expressed (OS of 18 months vs 29 months, $p = 0.0148$, Fig. 6e). A poor IMDC score was indicative of shorter PFS and OS (Table 4). The levels of Plk1 and the IMDC score were analyzed in a multivariate Cox regression model on PFS. Plk1 mRNA levels are indicative of PFS for patients on TKI independently of the IMDC score ($p = 0.006$, Table 4). Hence, Plk1 expression may participate in the therapeutic decision in addition to clinical parameters. Plk1 expression was more important in ccrcc1&4 as compared to ccrcc2&3 molecular subtypes of patients (Fig. 6f). This higher expression is relevant since patients of the ccrcc1&4 subgroup are in

therapeutic impasses. SETD2 mutations are mostly present in the ccrcc1 subtype (Supplementary Figure S7 in[11]). Thus, SETD2 mutations and Plk1 appear as frequently represented in ccrcc1-subtype tumors. Hence, patients of this subtype that respond poorly to TKIs and immunotherapy, are good candidates for Plk1 targeting agents such as volasertib.

## Discussion

Plk1 is upregulated in fast growing tumors. However, its regulation in cancer cells is poorly understood. Fast growing tumors often experience hypoxia and nutrient deprivation because of insufficient blood supply. These stress conditions result in cell senescence or cell death. However, this challenging situation induces the selection of more aggressive cancer stem cells that are resistant to chemotherapies[25]. How tumor cells can still proliferate and acquire invasive properties in severe hypoxia? At this time of writing, this interesting phenomenon cannot be explained. However, several hypoxia-related characteristics are associated with fast tumor growth, including: (1) necrosis linked to deprivation of oxygen; (2) hypoxia-dependent VEGF induction and subsequent angiogenesis; (3) poor delivery of cancer drugs leading to their inefficiency; (4) hypoxia-dependent upregulation of several growth factors and cytokines; (5) tumor microenvironment shaping; and (6) adaptation of cancer cell metabolism. In this study, we provide an example of a hypoxia-regulated oncogenic protein, Plk1, which significantly contributes to cancer metastasis and drug responses. This study more specifically addresses the role of Plk1 is ccRCC but we believe that these results can be generalized to several tumors.

Most hypoxia-regulated genes are under the control of HIF-1, which is a master regulator of hypoxia-triggered responses. Hypoxia-induced VEGF and EPO expression is an example of gene controlled by HIF-1[26]. In addition to environmental hypoxia, mutations also trigger a similar hypoxia-like response in cancer cells. In this report, we focused on ccRCC that often carry a dysfunctional VHL. In the absence of VHL, HIF upregulates the transcription of target genes independently of the oxygen concentration. Although HIF-1 and HIF-2 bind to similar sequences, HIF-1 drives genes involved in metabolism, whereas HIF-2 stimulates genes coding for pro-survival factors (30). Therefore, HIF-1α is considered as a tumor suppressor whereas HIF-2α is considered as an oncogene in ccRCC (31,32). We showed that hypoxia-dependent upregulation of Plk1 depends on increased transcription dependent on HIF-2 but not HIF1, and on mutation

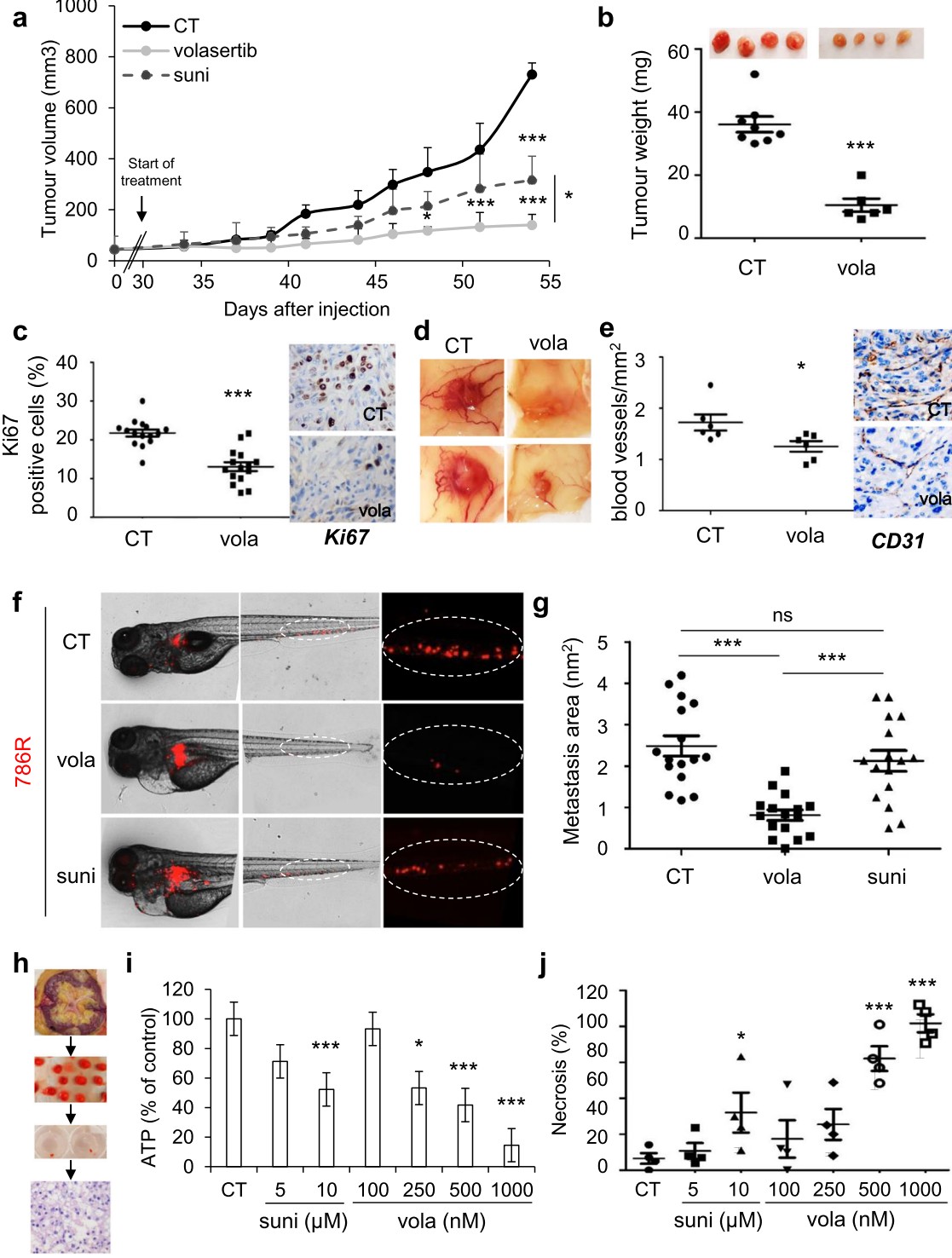

**Fig. 5 Volasertib inhibited the growth of experimental ccRCC in mice, decreased metastasis in the zebrafish model, and induced the death of 3D ccRCC primary tumors. a–e** 7.10⁶ 786 cells were subcutaneously injected into nude mice ($n = 8$ per group). 30 days after injection, all mice developed tumors and were treated with the control solution, or 25 mg/kg volasertib (vola) by gavage twice a week, or 40 mg/kg sunitinib by gavage five times a week. **a** The tumor volume was measured twice a week as described in supplementary Materials and methods. **b** The tumor weight at the end of the experiment. **c** IHC of KI67 (proliferative cells). Representative images are shown. **d** Representative images of tumors with blood vessels are shown. **e** IHC of CD31 (blood vessels). Representative images are shown. **f**, **g** Zebrafish embryos ($n = 45$) were injected with 786 R (labeled with red DiD) into the perivitelline space; 24 h later, only those with metastasis are chosen and treated for 48 h with sunitinib (suni, 1 μM) or vola (50 nM). Zebrafish embryos were monitored for investigating tumor metastasis using a fluorescent microscope. Representative images are shown (**f**) and the area of metastasis is quantified (**g**). **h–j** A sample of tumors following nephrectomy of the patient was analyzed by a pathologist (4 ccRCC patients). The tumor sample was then cut into fragments of about 5 mm³, cultured in a specific medium, and treated for 72 h with sunitinib (suni) or vola (**h**). The concentration of ATP provides a read-out of the tumor fragment viability (**i**). Tumor fragments were paraffin-embedded and stained with HES to quantify the areas of necrosis (**j**). Statistics were determined using an unpaired Student's *t* test (**a**, **b**, **c**, **e**) or ANOVA analysis (Bonferroni's comparison, **g**, **h**, **j**): *$p < 0.05$, **$p < 0.01$, ***$p < 0.001$.

**Table 3 The characteristics of the metastatic ccRCC patients included in the study.**

|  | Total | Low Plk1 | High Plk1 | *p* value |
|---|---|---|---|---|
| Number | 158 | 117 | 41 | |
| pM | | | | |
| 1 | 158 (100%) | 43 (100%) | 15 (100%) | |
| Fuhrman grade | | | | 0.008 |
| 2 | 4 (2.5%) | 2 (1.7%) | 2 (4.9%) | |
| 3 | 53 (33.7%) | 47 (40.5%) | 6 (14.6%) | |
| 4 | 100 (64.7%) | 67 (57.8%) | 33 (80.5%) | |
| Heng score (IMDC score) | | | | |
| Good | 17 (10.9%) | 15 (14.2%) | 2 (4.9%) | 0.251 |
| Intermediate | 103 (66%) | 67 (63.2%) | 27 (65.8%) | |
| Poor | 36 (23.1%) | 24 (22.6%) | 12 (29.3%) | |
| Molecular subtype | | | | 1.4E-11 |
| ccrcc1 | 38 (25.5%) | 29 (26.4%) | 9 (23%) | |
| ccrcc2 | 72 (48.3%) | 67 (60.9%) | 5 (12.9%) | |
| ccrcc3 | 9 (6%) | 7 (6.35%) | 2 (5.1%) | |
| ccrcc4 | 30 (20.1%) | 7 (6.35%) | 23 (59%) | |
| First-line treatment | | | | 0.277 |
| Sunitinib | 91 (57.6%) | 68 (58.1%) | 23 (56.1%) | |
| Pazopanib | 53 (33.5%) | 37 (31.6%) | 16 (39%) | |
| Sorafenib | 14 (8.9%) | 12 (10.3%) | 2 (4.9%) | |
| PFS (months) / Progression (%) | 8/ (82%) | 11/ (79%) | 4/ (93%) | 0.0015 |
| OS (months) / Death (%) | 26/ (79%) | 29/ (80%) | 18/ (93%) | 0.0148 |

Patient characteristics and univariate analysis with the Fisher's or $\chi^2$ test. Statistical significance (*p* values) is indicated.

of SETD2 in human ccRCC. HIF-2 regulates the expression of several growth factors and cytokines, which collaboratively promote oncogenesis in a Plk1-dependent or independent manner. At an advanced stage of tumor development, the hypoxia-like response triggered by genetic alterations and/or environmental hypoxia play a dual role in driving cancer progression. We provide clinical evidences supporting this hypothesis by showing that high Plk1 expression in hypoxic/necrotic zones are correlated to a poor prognosis. Hence, Plk1 appears as a central player facilitating tumor development and progression.

Metastatic ccRCC patients relapse despite angiogenesis (VEGFR-TKI) and immune checkpoint inhibitors. Predictive markers of efficacy of the current treatments and new therapeutic targets are urgently needed for patients in therapeutic impasses. In addition to its anti-angiogenic effects on endothelial cells, sunitinib, one of the current reference treatment for metastatic patients, directly targets tumor cells to inhibit their proliferation, migration, and survival. ccRCC patients receiving sunitinib ineluctably develop resistance through a genetic adaptation of tumor cells leading to their survival in the presence of the drug. Sunitinib-resistant ccRCC cells exhibit higher Plk1 expression, suggesting that Plk1 induction is part of a genetic program associated with resistance to sunitinib. Thus, Plk1 blockade represents an attractive and alternative therapeutic solution for treating sunitinib-resistant ccRCC patients. Indeed, we provide compelling evidences showing that sunitinib-resistant ccRCC cells are highly sensitive to Plk1 inhibition. This exciting finding warrants clinical validation.

Plk1-dependent resistance to sunitinib relies on the inhibition of p53-dependent senescence[21]. The majority of ccRCC cells do not have a p53 mutation. Over-expression of Plk1 blocked p53 activation, resulting in the inhibition of sunitinib-induced senescence. In prostate cancer, olaparib (PARP inhibitor) treatment caused accumulation of cells in G2/M resulting in increased Plk1 expression and resistance[27]. An imbalance in the cross-talk between Plk1 and p53 results in the deregulation of oncogenic pathways contributing in cancer development and resistance to treatment.

Epithelial-to-mesenchymal transition (EMT) enables cancer cells to avoid apoptosis, anoïkis, and oncogene addiction[28]. Over-expression of Plk1 increases cell migration, a key process induced during EMT. Our results are consistent with the downregulation of epithelial and upregulation of mesenchymal markers in prostate epithelial cells over-expressing Plk1[29].

Plk1 is associated with resistance to doxorubicin, paclitaxel, and gemcitabine[30]. Targeting the addiction to Plk1 appears relevant to increase the sensitivity to chemotherapy[14,15], which is consistent with volasertib-dependent ccRCC cell death in sunitinib-sensitive and sunitinib-resistant cells.

The relevance of Plk1 targeting with a non-clinically approved Plk1 inhibitor (BI 25366, intratumor injection, irrelevant for cancer patients) for ccRCC was described by an independent approach (gene expression profiling and RNAi screening)[31]. However, to our knowledge, this research paper of Ding et al. was the only one describing the role of Plk1 in ccRCC aggressiveness. One part of our study confirms this important result on conventional cell lines from the ATCC as described by Ding et al. The importance of Plk1 in cell lines constitutes only one small part of our manuscript confirming the results of Ding et al. To go further, we showed the efficacy of Plk1 targeting on surgically resected tumors of patients which is robust, added value to this general concept. These experiments are key to convince clinicians about the relevance to test Plk1 inhibitors in early-phase clinical trials, especially for patients experiencing relapses on angiogenic or immune checkpoint inhibitors. We are currently discussing this opportunity with oncologists of the cancer center of Nice, FRANCE.

We further explored the relation between over-expression of Plk1 and ccRCC aggressiveness. The link between two major cancer hallmarks, cell proliferation through activation of Plk1 and hypoxia through HIF-α stabilization, constitute an important mechanistic part of the present study. ccRCC represents a

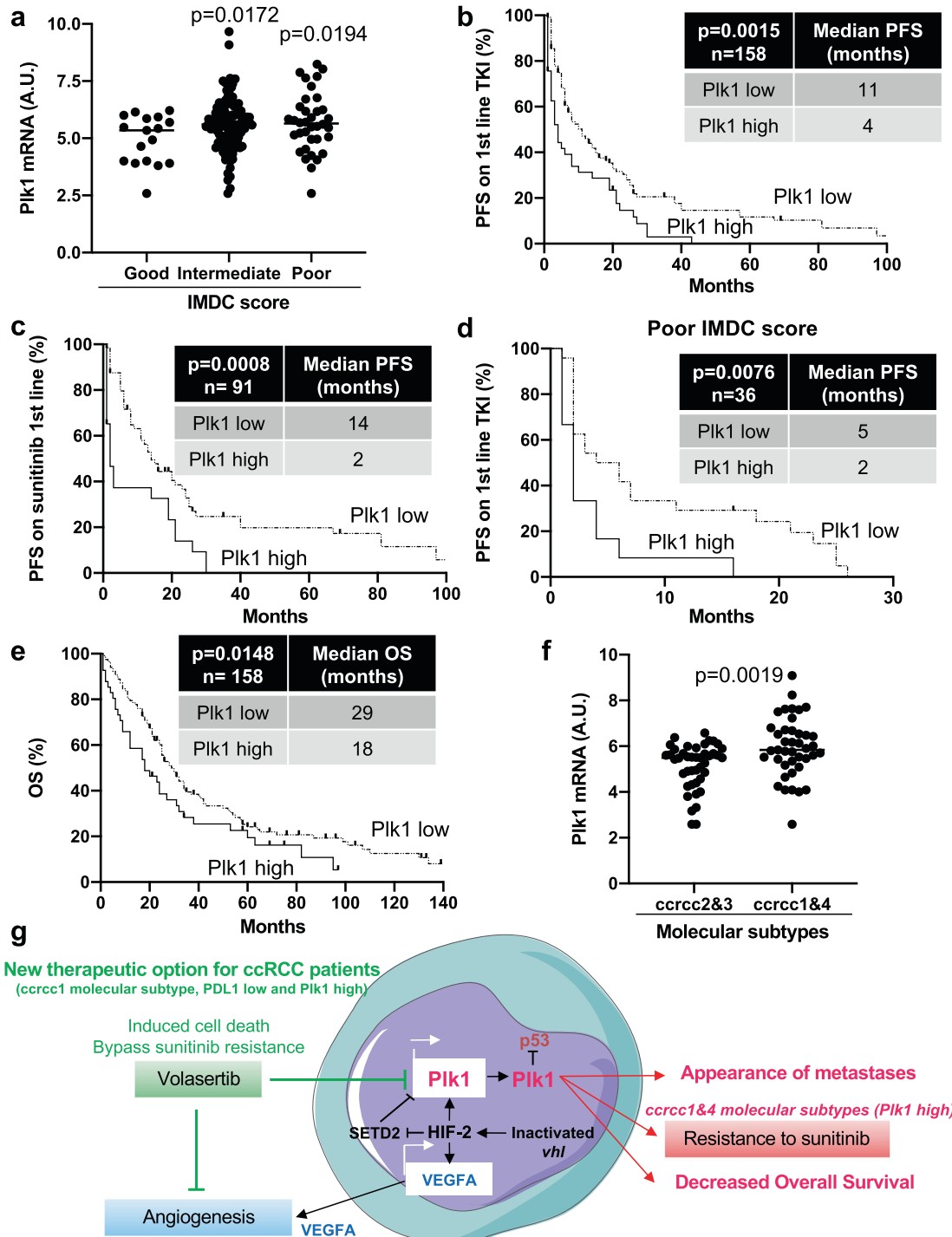

**Fig. 6 Plk1 is associated with resistance to fist-line VEGFR-TKI treatment in ccRCC.** The tumors of 58 metastatic ccRCC patients were analyzed for the Plk1 mRNA level. **a** The levels of Plk1 mRNA in tumors from patients with a good, intermediate, and poor IMDC score were compared. **b–e** The levels of Plk1 mRNA in tumors of 58 metastatic ccRCC patients treated with VEGFR-TKI in the first line correlated with PFS (**b**) or with OS (**e**). The levels of Plk1 mRNA in tumors of 27 metastatic ccRCC patients treated with sunitinib in the first line correlated with PFS (**c**). The levels of Plk1 mRNA in tumors from patients with an intermediate IMDC score correlated with PFS (**d**). **f** The levels of Plk1 mRNA in tumors of the ccrcc2&3 and ccrcc1&4 subtypes were compared. The third quartile of Plk1 expression was chosen as the cut-off value. Statistics were determined using an unpaired Student's t test. The Kaplan-Meier method was used to produce survival curves and analyses of censored data were performed using Cox models. Statistical significance (p values) is indicated (see Table 3). **g** The summary diagram of our study describing the phenomenon linking a physical driver of tumor aggressiveness (hypoxia) to a biological determinant of tumor cell proliferation and angiogenesis (Plk1). The link between the two actors is HIF-2, which drives *Plk1* gene transcription through a SETD2-dependent mechanism. In addition to its tumor promoting role, Plk1 drives resistance to TKI and appears as a key target for a subgroup of metastatic ccRCC patients in therapeutic impasses.

**Table 4 Multivariate analysis — metastatic ccRCC patients.**

| IMDC score | Description | Median (months) | Coef | HR | *p* value |
|---|---|---|---|---|---|
| PFS | Good | 27 | 0.654 | 1.923 | 0.0368 |
| | Intermediate | 9 | 1.38 | 3.974 | 0.00007 |
| | Poor | 3 | | | |
| OS | Good | 101 | 1.136 | 3.114 | 0.004 |
| | Intermediate | 26 | 1.969 | 7.16 | 0.00002 |
| | Poor | 11.5 | | | |

| PFS first-line TKI | Description | | Coef | HR | *p* value |
|---|---|---|---|---|---|
| Biological parameter Plk1 | | | 0.54 | 1.716 | 0.006 |
| Clinical parameters IMDC score | Good | | 0.599 | 1.821 | 0.06 |
| | Intermediate | | 1.343 | 3.832 | 0.001 |
| | Poor | | | | |

Multivariate analysis of the IMDC score and PFS or OS, and multivariate analysis of Plk1, the IMDC score, and PFS. The multivariate analysis was performed using Cox regression adjusted to the IMDC score. Statistical significance (*p* values) is indicated (see Fig.6).

paradigm for HIF-dependent tumor aggressiveness. The generalization of this concept to different tumors (acute myeloid leukemia, adrenocortical, bladder, breast, cervical, colorectal, esophageal, head and neck, liver, lung, pancreatic, ovarian, prostate, thyroid, and testicular cancers, and glioblastoma, glioma, mesothelioma, melanoma, and sarcoma) brings an added value to improve the treatment of these cancers.

Targeting Plk1 inhibited tumor and endothelial cell proliferation in mice models. Therefore, the therapeutic efficacy of Plk1 inhibitors also relies on the inhibition of angiogenesis, a key phenomenon in ccRCC. We also showed that aggressive ccRCC cells metastasized in zebrafish tails without genetic modification beforehand, and Plk1 inhibition prevented this metastatic spreading. Hence, the zebrafish model represents a relevant model to test new therapeutic drugs as recently reviewed[32].

Plk1 is a driver of tumor growth orchestrated by the HIF-2 oncogenic pathway. Our study linked Plk1 to a shorter survival in both M0 and M1 patients. Plk1 is a prognostic factor independent of IMDC score. Hence, a biological marker independent of clinical parameters provides an added value to the management of patients.

The gold-standards for metastatic ccRCC patients in the first-line are TKI, immunotherapy (anti-PD1 + anti-CTLA4[33]), or a combination of both therapies[34,35].

The clinical parameters of the IMDC score are poorly informative for patients in the intermediate group. The four subtypes ccrcc1–4, based on biological parameters, refine the therapeutic strategy[11]. ccrcc1-tumors have an immune-cold phenotype indicative of immunotherapy refractoriness and a poor response to TKI. Tumors of the ccrcc1-subtype strongly express Plk1. High Plk1 mRNA levels correlated with a poor response to immunotherapy[36]. Therefore, Plk1 inhibition represents a relevant strategy for these patients. The following nomogram appears decisional for the therapeutic strategy for patients of the different subgroups:

ccrcc2&3 subtypes (low PDL1 and Plk1 expression) are eligible for TKI,

ccrcc4 subtype (high PDL1 and Plk1 expression) are eligible for immunotherapy,

ccrcc1 subtype (low PDL1 expression but strong Plk1 expression) are eligible for treatment with Plk1 inhibitors

Our study deciphered the phenomenon linking a physical driver of tumor aggressiveness (hypoxia) to a biological determinant of tumor cell proliferation and angiogenesis (Plk1). The link between the two actors is HIF-2, which drives *Plk1* gene transcription through a SETD2-dependent mechanism. In addition to its tumor promoting role, Plk1 drives resistance to TKI and appears as a key target for a subgroup of metastatic ccRCC patients in therapeutic impasses (Fig. 6g).

## Methods

**Reagents and antibodies**. Inhibitors were purchased from Selleckchem. Anti-HSP90 (ref: sc-515081, dilution: 1/1000) and anti-tubulin (ref: sc-5286, dilution: 1/1000) antibodies were purchased from Santa Cruz Biotechnology. Anti-Plk1 (ref: ab19777, dilution: 1/1000) antibodies were purchased from Abcam. The anti-HIF-2α (ref: NB100-122, dilution: 1/1000) antibody was purchased from Novus Biochemicals. The rabbit polyclonal anti-HIF-1α antibodies (home-made antiserum 2087, dilution: 1/1000) were previously described[37].

**Cell culture**. Different kidney cancer cell lines and primary cells were used in this study, including clear cell RCC (ccRCC) cell lines (caki-2, RCC4, RCC10, 786-O) and primary ccRCC cells (CC, MM), and also other types of RCC including papillary (pRCC) cell lines (A-498, ACHN) and primary RCC cells with a TFE3 translocation (TF). RCC4 (R4), ACHN (A), Caki-2 (C2), 786-O (786), and A498 (498) RCC cell lines were purchased from the American Tissue Culture Collection. RCC10 (R10) was a kind gift from Dr. W.H. Kaelin (Dana-Farber Cancer Institute, Boston, MA). Primary cells were previously described,[19] and they were cultured in a medium specific for renal cells (PromoCell, Heidelberg Germany); 786R and RCC10R cells were previously described[38]. An InvivO$_2$ 200 anaerobic workstation (Ruskinn Technology Biotrace International Plc) set at 1% oxygen, 94% nitrogen, and 5% carbon dioxide was used for hypoxic conditions[39].

## Patients

*Ethics approval.* These studies were approved by the ethics committee at each participating center and run in agreement with the International Conference on Harmonization of Good Clinical Practice Guideline. Informed consent was obtained from all individual participants included in the study. All patients gave written consent for the use of tumor samples for research. The study included only the adult patients. This study was conducted in accordance with the Declaration of Helsinki.

*TCGA cohort: non-metastatic (M0) and metastatic (M1) ccRCC patients — Plk1 mRNA analysis.* Normalized RNA sequencing (RNA-Seq) data produced by The Cancer Genome Atlas (TCGA) were downloaded from cBioportal (www. cbioportal.org, TCGA Provisional; RNA-Seq V2). See supplementary Materials and methods, Supplementary Figs. 2 and 9.

OS was calculated from patient subgroups with Plk1 mRNA levels (*z*-score) that were lesser or greater than the third quartile value.

*French cohort: M0 ccRCC patients — Plk1 mRNA analysis.* Primary tumor samples of M0 ccRCC patients were obtained from the Rennes[40] and Bordeaux University Hospitals and from the UroCCR group (Fig. 1 and Table 1). Disease-free survival (DFS) and OS were calculated from patient subgroups with Plk1 mRNA levels that were lesser or greater than the third quartile value.

*Tissue microarray (TMA) cohort: M0 and M1 ccRCC patients — Plk1 protein analysis.* TMAs of primary tumor samples of ccRCC patients were obtained from the Bordeaux University Hospital. The Plk1 score was calculated from the percentage of labeled cells (0–100%) multiplied by the staining intensity (0–3). The DFS, PFS, and OS were calculated from patient subgroups with Plk1 expression that was lesser or greater than the third quartile (score = 120) IHC score (Supplementary Fig. S3 and Supplementary Table 2).

*Belgium cohort: M1 ccRCC patients — Plk1 mRNA analysis*. Samples of primary tumors from metastatic (M1) ccRCC patients were obtained from Leuven Hospital (Fig. 6 and Table 3). Plk1 mRNA levels were measured using a customized Nanostring Counter(c) gene panel.

**siRNA assay**. siRNA transfection was performed using Lipofectamine RNAiMAX (Invitrogen). Cells were transfected with either 50 nM of siHIF-1α (siH1)[41] and/or HIF-2α (siH2)[42] or sicontrol (siCT, Ambion, 4390843). Two days later, experiments were performed as described above.

**Quantitative real-time PCR (qPCR) experiments**. Of the total RNA, 1 µg was used for the reverse transcription, using the QuantiTect Reverse Transcription kit (QIAGEN), with a blend of oligo (dT) and random primers to prime first-strand synthesis. SYBR master mix plus (Eurogentec) was used for qPCR. The mRNA level was normalized to 36B4 mRNA.

**Luciferase assays**. Transient transfections were performed using 2 µl of lipofectamine (GIBCO BRL) and 0.5 µg of total plasmid DNA-*Renilla luciferase* in a 500 µl final volume. The firefly control plasmid was co-transfected with the test plasmids to control for the transfection efficiency; 24 h after transfection, cell lysates were tested for renilla and firefly luciferase. All transfections were repeated four times using different plasmid preparations. LightSwitch™ Promoter Reporter Plk1 was purchased from Active motif.

**Cell viability (XTT)**. Cells ($5 \times 10^3$ cells/100 µl) were incubated in a 96-well plate with different effectors for the times indicated in the figure legends. To each well, 50 µl of sodium 3′-[1-phenylaminocarbonyl)-3,4- tetrazolium]-bis(4-methoxy-6-nitro) benzene sulfonic acid hydrate (XTT) reagent was added. The assay is based on the cleavage of the yellow tetrazolium salt XTT to form an orange formazan dye by metabolically active cells. Absorbance of the formazan product, reflecting cell viability, was measured at 490 nm. Each assay was performed in quadruplicate.

**Chromatin immunoprecipitation (ChIP)**. ChIP experiments were performed as already described[43]. Briefly, cells were grown in normoxia or hypoxia (1% $O_2$) for 24 h ($5–10 \times 10^6$ cells were used per condition). Cells were then fixed with 1% (v/v) formaldehyde (final concentration) for 10 min at 37 °C and the action of the formaldehyde was stopped by the addition of 125 mM glycine (final concentration). Next, cells were washed in cold PBS containing a protease inhibitor cocktail (Roche), scrapped into the same buffer and centrifuged. The pellets were resuspended in lysis buffer, incubated on ice for 10 min, and sonicated to shear the DNA into fragments of between 200 and 1000 base pairs. Insoluble material was removed by centrifugation and the supernatant was diluted 10-fold by the addition of ChIP dilution buffer and pre-cleared by the addition of salmon sperm DNA/ protein A agarose 50% slurry for 1 h at 4 °C. About 5% of the diluted samples was stored and constituted the input material. Immunoprecipitation was then performed by the addition of anti-HIF-2α or anti-tubulin as IgG control antibodies for 24 h at 4 °C. Immune-complexes were recovered by adding 50% of salmon sperm DNA/protein A agarose and washed sequentially with low-salt buffer, high-salt buffer, LiCl buffer, and TE. DNA complexes were extracted in elution buffer, and cross-linking was reversed by incubating overnight at 65 °C in the presence of 200 mM NaCl (final concentration). Proteins were removed by incubating for 2 h at 42 °C with proteinase K and the DNA was extracted with phenol/chloroform and precipitated with ethanol. Immunoprecipitated DNA was amplified by PCR with the following primers:

Plk1 primers: Forward: 5′-AGTGAACCGCAGGAGCTTTC-3′, Reverse: 5′-TT AAAATCCAAACCCGCCCG-3′;

Positive control (PDH3) primers: Forward: 5′-TTCTCTGGTGACTGGGGTAG AGAT-3′, Reverse: 5′-GAGCCCATGCAATTAGGCACAGTA-3′;

Negative control (Ang2 – 9351) primers: Forward: 5′-TCACCTGAGGATACA GAGAC-3′, Reverse: 5′-AGCGACAGGCAAATCTATCCA-3′.

**Cytospin preparations and hematoxylin-eosin staining**. Cytospin preparations were also obtained using the cytocentrifuge (Thermo Scientific Cytospin 4, Thermo, Pittsburgh, PA, U.S.A.) at 900 rpm for 9 min. Smears were HE stained for morphological assessment.

**Measurement of the caspase activity**. After stimulation, cells were lysed for 30 min at 4 °C in lysis buffer[44], and lysates were cleared at 10,000*g* for 15 min at 4 °C. Each assay was done with 25 µg of protein. Cellular extracts were incubated in a 96-well plate with Ac-DEVD-AMC (caspase 3) or Ac-VDVAD-AMC (caspase 2) for various times at 37 °C. Caspase activity was measured at 410 nm in the presence or absence of 1 µM of Ac-DEVD-CHO.

**Flow cytometry**
*Analysis of apoptosis*. After stimulation, cells were washed with ice-cold PBS and stained with the Annexin-V-FLUOS Staining Kit (Roche) according to the manufacturer's instructions. Fluorescence was measured using the FL2 and FL3

channels of a fluorescence-activated cell sorter apparatus (FACS-Calibur cytometer).

*Cell cycle analysis*. After treatment, cells were washed, fixed in ethanol 70%, and, finally, left overnight at −20 °C. Next, cells were incubated in PBS, 3 µg/ml RNase A, and 40 µg/ml propidium iodide (PI) for 30 min at 4 °C. Cellular distribution across the different phases of the cell cycle or DNA content was analyzed with a FACS-Calibur cytometer.

**Senescence evaluation**. After treatment, cells were washed and stained with the Senescence β-Galactosidase Staining Kit (CST) according to the manufacturer's instructions. β-Galactosidase activity at pH 6 is a known characteristic of senescent cells.

**Tumor xenograft experiments**. *Ectopic model of ccRCC:* $7 \times 10^6$ 786 cells were injected subcutaneously into the flank of 5-week-old nude (nu/nu) female mice (Janvier, France). The tumor volume was determined with a caliper ($v = L*l^2*0.5$). When the tumor reached 50 mm$^3$, mice were treated five times a week for 4 weeks, by gavage with placebo (dextrose water vehicle), sunitinib (40 mg/kg) or twice a week for 4 weeks with volasertib (25 mg/kg). This study was carried out in strict accordance with the recommendations of the Guide for the Care and Use of Laboratory Animals. Our experiments were approved by the "Comité national institutionnel d'éthique pour l'animal de laboratoire (CIEPAL)" (reference: NCE/ 2015-255).

**Immunohistochemistry**. Sections from blocks of formaldehyde-fixed and paraffin-embedded tumors were examined for immunostaining. Sections were incubated with monoclonal anti-mouse CD31 (clone MEC 13.3, BD Pharmingen, diluted at 1:500) or Ki67 (clone MIB1, DAKO, Ready to use) antibodies. Biotinylated secondary antibody (DAKO) was applied and binding was detected with the substrate diaminobenzidine against a hematoxylin counterstain.

**Zebrafish metastatic tumor model**. All animal experiments were approved by the Northern Stockholm Experimental Animal Ethical Committee. Zebrafish embryos were raised at 28 °C under standard experimental conditions. At the age of 24 hpf, they were incubated in aquarium water containing 0.2 mmol/l 1-phenyl-2-thiourea (PTU, Sigma). At 48 hpf, they were dechorionated with a pair of sharp-tip forceps and anesthetized with 0.04 mg/ml Tricaine (MS-222, Sigma). Anesthetized embryos were subjected for microinjection; 786R tumor cells were labeled in vitro with a Vybrant DiD cell-labeling solution (LifeTechnologies). They were resuspended in PBS and approximately 5 nl of the cell solution was injected in the perivitelline space (PVS) of each embryo by an Eppendorf microinjector (FemtoJet 5247). Non-filamentous borosilicate glass capillary needles were used for injection and the injected zebrafish embryos were immediately transferred into PTU aquarium water with treatment; 24 h later, only zebrafishes with metastases were treated. Metastases were monitored for 48 h using a fluorescent microscope (Nikon Eclipse 90).

**Treatment of primary ccRCC tumors**. Four ccRCC samples obtained just after nephrectomy were provided thorough a collaboration with the Nice University Hospital. Tumor samples were then cut into pieces of about 5 mm$^3$, cultured in a specific medium[19], and treated for 72 h with sunitinib or volasertib. Tumor fragments were then paraffin-embedded and analyzed using HES for the quantification of necrotic areas. ATP concentration in the lysed tumor fragments represented a read-out of their viability.

**Statistics and reproducibility**
*For in vitro and in vivo analysis*. Results are represented as means of three or more independent experiments (biological replication). All data are expressed as mean ± standard error of mean (SEM). Statistical significance and *p* values were determined by the two-tailed Student's *t* test. One-way ANOVA was used for statistical comparisons. Data were analyzed with Prism 5.0b (GraphPad Software) by one-way ANOVA with Bonferroni post hoc test.

*For patients*. The Student's *t* test was used to compare continuous variables, and chi-square test or Fisher's exact test (when the conditions for use of the $\chi^2$-test were not fulfilled) was used for categorical variables. To guarantee the independence of Plk1 as a prognostic factor, the multivariate analysis was performed using a Cox regression adjusted to the Fuhrman grade. DFS was defined as the time from surgery to the appearance of metastasis. PFS was defined as the time between surgery and progression, or death from any cause, censoring those alive and progression-free at the last follow-up. OS was defined as the time between surgery and the date of death from any cause, censoring those alive at the last follow-up. The Kaplan-Meier method was used to produce survival curves and analyses of censored data were performed using Cox models. All analyses were performed using R software, version 3.2.2 (Vienna, Austria, https://www.r-project.org/).

**Reporting summary**. Further information on research design is available in the Nature Research Reporting Summary linked to this article.

## Data availability

All data generated or analyzed during this study are included in this published article (and its supplementary information files). Source Data are available in Supplementary Data 1.

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

## Acknowledgements

This work was supported by the French Association for Cancer Research (ARC), the Fondation de France, the Ligue Nationale contre le Cancer (Equipe Labellisée 2019) the French National Institute for Cancer Research (INCA), the National Agency for Research (ANR) and the FX Mora and Flavien Foundations. This study was conducted as part of the Centre Scientifique de Monaco Research Program, funded by the Government of the Principality of Monaco. The samples from Bordeaux and associated data were collected, selected, and made available within the framework of the clinicobiological project, National Cancer Database Kidney UroCCR supported by l'Institut National du Cancer (INCa). We thank the IRCAN core facilities (animal and cytometry facilities) for technical help. We thank also the Department of Pathology, especially Arnaud Borderie and Sandrine Destree, for technical help.

## Author contributions

Investigation, M.D., A.V., L.S.C., P.D.N., X.H., N.N., W.S., A.H., S.T., J.P., A.B., D.A., and J.P.; Methodology, M.D.; Resources; A.V., N.M.M., B.M., N.R.L., K.B., A.R., P.A., J.Z.R., S.G., D.B., B.B., Y.C., J.C.B., and D.A.; Conceptualization, M.D., D.A., and G.P.; Statistical analysis, J.V., R.S., and E.C.; Writing-original draft, Y.C., M.D., and G.P.; Supervision, Project administration, and funding acquisition, M.D. and G.P.

## Competing interests

The authors declare no competing interests.
