## [Peer Review File · Communications Biology]

Reviewers' comments:

Reviewer #1 (Remarks to the Author):

The manuscript presents data on the regulation of PLK1 by HIF2 and its implications for clear cell renal cell carcinoma. The data shows that PLK1 expression increases under hypoxia or VHL inactivation and that HIF2 appears to contribute to this increase. On balance, the data is solid and reasonably interpreted by the authors. However, the data is mostly associative observations, and could benefit from either clearer explanation or some simple additional data. There is one overarching concern, namely that it might not just be HIF-2 α that could be the reason for the changes in PLK1 expression they are seeing. The key experiments are in Figure 2 F, G, H, and I, in which they are knocking down HIF2 or expressing VHL, and the resulting changes in PLK1 expression. The effect of VHL overexpression was quite modest. Moreover, PLK1 levels are quite well known to be cell cycle regulated. In fact, PLK1 is nearly as much of a marker as proliferating cells as Ki67 in Fig 5C. The extra explanation or data needed is whether HIF2 knockdown or VHL overexpression altered the proliferation of the cells, and hence the changes in PLK1 expression might have been due to less direct effects on the cell cycle. Related, it would have been interesting to see PLK1 expression in cells grown in hypoxic conditions but also knocked down for HIF2 (figure 2H). If the effects of altered proliferation status of the models could be clarified, the manuscript provides interesting data with implications for renal cell carcinoma that could be clinically actionable.

Minor concerns.

The claim that volasertib had no effects on the normal 15S cells (Supplementary Fig 7 B-D) is only thinly supported by the data shown. There is no colony forming data for this cell line in A, and the flow cytometry data in B-D is at a 48 h time point, which in a more slowly proliferating cell line might not be long enough to see an induction of apoptotic cells.

The manuscript is well written overall but contains a few awkward phrases throughout. Just a few examples.

Line 95, they propose the term theranostic, but this phrase is already widely used (>10K hits in pubmed) for imaging/therapy applications.

Line 172, well-admitted. Can just say "using luciferase reporter assays."

Line 244, patients baring to large tumors. patients bearing large tumors.

Line 337 and the following sentences. The manuscript claims that tumors can proliferate at a high rate in severe hypoxia and proposes seven different possibilities, but this goes against the general dogma that the bulk of tumor cells in the hypoxic core of a solid tumor are senescent or dying. This also doesn't match up with the experimental emphasis on metastasis. If this is a phenomenon more specific to renal cell carcinoma, this part of the discussion would be better served to be written in a more focused paragraph.

Reviewer #2 (Remarks to the Author):

Authors showed compelling evidence about the hypoxia dependent PLK1 mediated drug resistance and metastasis in clear cell renal cell carcinoma (ccRCC). HIF-2 (hypoxia-inducible factor-2) was stabilized by hypoxia and the HIF-2 upregulated PLK1 levels in normal primary cells and ccRCC cells. In addition to that, genetic defect in VHL gene induced hypoxia like response in ccRCC tumor which was supported by the analysis of VHL gene in French cohort of ccRCC patient's tumors. 43 out of 111 (38.7 %) of the patients' tumor had two alleles of the VHL gene were either deleted or mutated or promoter was methylated. Approximately 80% of ccRCC patients' tumor harbor dysfunctional VHL. Tumors with inactivated two alleles of VHL and/or methylated VHL promoter presented higher PIK1 mRNA levels as compared to tumors with normal or with only one inactivated VHL allele. This showed that the VHL activity negatively correlates with PLK1 expression.

Sunitinib-resistant ccRCC cells exhibited higher PLK1 expression and those cells showed high sensitivity to volasertib. Studies showed PLK1 involve in drug resistance. I.e study by Zhu et al., (Cancer Sci. 2013, PMID: 23578198) showed PLK1 dependent resistance to sunitinib depended on inhibition of p53 mediated senescence in ccRCC. Moreover, study by Li et al., (Mol Cancer Ther, 2017, PMID: 28069876) showed PLK1 involved in olaparib (PARP inhibitor) resistance in p53 mutant castration resistance prostate cancer cells. Olaparib treatment clearly caused accumulation of cells at G2/M thus elevation of PLK1, which eventually counteracted the efficacy of Olaparib.

Specific points to be addressed:

1. ccRCC (R10) and ccRCC sunitinib resistant cells (R10R) were treated with BI6727 (100 nM) for cell viability, apoptosis, and H&E experiments (Figure 4 G to I). Results about the apoptosis, cell viability and mitotic phenotype in BI6727 treated ccRCC agree with PLK1 inhibition data published before with other cancer cells line. PLK1 inhibitor at that dose may not just inhibit PLK1 but also other kinases. Generally, it is common practice to apply pharmacological as well as genetic approaches to study loss of function of protein. So, I recommend to use another approach to inhibit PLK1 via i.e siRNA or shRNA in their experiments (Cell viability, apoptosis assay and H&E staining).

2. They did not address how PLK1 reversed chemoresistance to sunitinib. I recommend to do senescence Associated (SA) β -galactosidase assay to show how PLK1 bypassed senescence as mechanism of chemo sensitization.

3. They showed PLK1 was upregulated in 786R (resistant ccRCC cells) compared to 786 cells (ccRCC cell) (Figure 4F). I recommend to show the cell cycle profile of 786 and 786R (resistance cells) by flow cytometry. Also, I recommend mentioning about p53 status (mutant or WT) in those cells' lines (786 and 786R). Other study showed PLK1 involvement in Olaparib resistance in p53-mutant castration resistance prostate cancer cells. Olaparib treatment clearly causes accumulation of cells at G2/M thus elevation of Plk1, which eventually counteract the efficacy of olaparib.

Validity of any statistical analysis:

It is understandable that in some of their statistical analysis (i.e metastatic ccRCC patients-Belgium cohort Figure 6 B-D) patients number was low (39 to 58 patients). I do not have a big concern on it but would make the result more valid if they have more patients in their analysis (it could be possible that they did not have more patients).

Point by point answer to the reviewers

Reviewer #1:

The manuscript presents data on the regulation of PLK1 by HIF2 and its implications for clear cell renal cell carcinoma. The data shows that PLK1 expression increases under hypoxia or VHL inactivation and that HIF2 appears to contribute to this increase. On balance, the data is solid and reasonably interpreted by the authors.

We thank the reviewer for these comments.

However, the data is mostly associative observations, and could benefit from either clearer explanation or some simple additional data. There is one overarching concern, namely that it might not just be HIF-2alpha that could be the reason for the changes in PLK1 expression they are seeing. The key experiments are in Figure 2 F, G, H, and I, in which they are knocking down HIF2 or expressing VHL, and the resulting changes in PLK1 expression.

Moreover, PLK1 levels are quite well known to be cell cycle regulated. In fact, PLK1 is nearly as much of a marker as proliferating cells as Ki67 in Fig 5C. The extra explanation or data needed is whether HIF2 knockdown or VHL overexpression altered the proliferation of the cells, and hence the changes in PLK1 expression might have been due to less direct effects on the cell cycle.

We thank the reviewer for this relevant comment. We evaluated the cell cycle of 786, 786 with VHL overexpression or HIF2 siRNA. No significant difference in cell cycle was observed between 786 and 786 overexpressing VHL (Fig. 1A). siRNA against HIF2 induced an accumulation of cells in G2M (Fig. 1B). Nevertheless, the same accumulation was found with a siRNA against Plk1 at 48h, which is followed by a strong induction of cell death at 96h (SubG1, Fig. 1C).

Figure 1. *A* The cell cycle of 786 and 786 overexpressing VHL (786+VHL) was analyzed by flow cytometry. *B* 786 cell were transfected with siRNA against HIF-2 α (siH2) for 48 h. Cell cycle was analyzed by flow cytometry. *C* 786 cells were transfected with siRNA against Plk1 for 48h or 96h. The cell cycle was analyzed by flow cytometry.

Related, it would have been interesting to see PLK1 expression in cells grown in hypoxic conditions but also knocked down for HIF2 (figure 2H). If the effects of altered proliferation status of the models could be clarified, the manuscript provides interesting data with implications for renal cell carcinoma that could be clinically actionable.

We thank the reviewer for this relevant comment. We evaluated the expression of Plk1 and the cell cycle of VHL WT cells (A) in hypoxic conditions, transfected with siCT or siHIF2. In A cells, hypoxia-induced overexpression of Plk1 was reversed by siRNA against HIF2 (Fig. 2A). This figure is **included in the revised version of the manuscript (Figure 2H)**. 24h hypoxia reduced slightly the percentage of cells in G2M with a small increase of the percentage of cells in S phase. siRNA against HIF2 reverted these effects on the cell cycle (Fig. 2B).

Figure 2. A and B VHL-WT RCC cells (A) were transfected with siRNA against HIF-2 α (siH2) for 24h and then were cultured in normoxia (Nx) or hypoxia 1% O₂ (Hx) for 24 h. A Plk1, HIF-2 α expression were evaluated by immunoblotting. HSP90 served as a loading control. Control conditions were considered as the reference value (1). B The cell cycle was evaluated by flow cytometry.

Minor concerns :

The claim that volasertib had no effects on the normal 15S cells (Supplementary Fig 7 B-D) is only thinly supported by the data shown. There is no colony forming data for this cell line in A, and the flow cytometry data in B-D is at a 48 h time point, which in a more slowly proliferating cell line might not be long enough to see an induction of apoptotic cells.

We thank the reviewer for these comments. 15S cells are a normal kidney cells and hardly form colonies. However, volasertib did not decrease of the number of 15S cell colonies while volasertib totally blocked the formation of 786 cell colonies (Fig. 3A). We also evaluated apoptotic cell death at 48h and 96h. Volasertib induced a strong apoptotic cell death at 48h and 96h in primary ccRCC cells (CC) while the death of 15S cells was low (Fig. 3B).

Figure 3. A RCC cells (786) and healthy renal cells (15S) were incubated in the presence of volasertib (vola, 10 to 250 nM) and stained with Giemsa blue after 10 days. **B** Primary RCC cells (CC) and healthy renal cells (15S) were treated with 500nM volasertib for 48h or 96h. Cell death was evaluated by flow cytometry. Cells were stained with PI and AV. Histograms show AV⁺/PI⁻ cells (apoptosis) and AV⁺/PI⁺ cells (post-apoptosis or another cell death)

The manuscript is well written overall but contains a few awkward phrases throughout. Just a few examples.

Line 95, they propose the term theranostic, but this phrase is already widely used (>10K hits in pubmed) for imaging/therapy applications.

Line 172, well-admitted. Can just say “using luciferase reporter assays.”

Line 244, patients baring to large tumors. patients bearing large tumors.

We apologize for these mistakes. We corrected them and performed a new editing of the manuscript.

Line 337 and the following sentences. The manuscript claims that tumors can proliferate at a high rate in severe hypoxia and proposes seven different possibilities, but this goes against the general dogma that the bulk of tumor cells in the hypoxic core of a solid tumor are senescent or dying. This also doesn't match up with the experimental emphasis on metastasis. If this is a phenomenon more specific to renal cell carcinoma, this part of the discussion would be better served to be written in a more focused paragraph.

This paragraph describes general observations. Cells at the proximity of the hypoxic zones can still proliferate although the majority of them undergo death of senescence when they are far from supplying blood vessels. If they can survive to these stress conditions, adapted cells are considered as much more aggressive. Several publications describe these features. Our results explain probably one phenomenon linked to the selection of more aggressive cells.

Reviewer #2:

Authors showed compelling evidence about the hypoxia dependent PLK1 mediated drug resistance and metastasis in clear cell renal cell carcinoma (ccRCC). HIF-2 (hypoxia-inducible factor-2) was stabilized by hypoxia and the HIF-2 upregulated PLK1 levels in normal primary cells and ccRCC cells. In addition to that, genetic defect in VHL gene induced hypoxia like response in ccRCC tumor which was supported by the analysis of VHL gene in French cohort of ccRCC patient's tumors. 43 out of 111 (38.7 %) of the patients' tumor had two alleles of the VHL gene were either deleted or mutated or promoter was methylated. Approximately 80% of ccRCC patients' tumor harbor dysfunctional VHL. Tumors with inactivated two alleles of VHL and/or methylated VHL promoter presented higher PIK1 mRNA levels as compared to tumors with normal or with only one inactivated VHL allele. This showed that the VHL activity negatively correlates with PLK1 expression.

Sunitinib-resistant ccRCC cells exhibited higher PLK1 expression and those cells showed high sensitivity to volasertib. Studies showed PLK1 involve in drug resistance. I.e study by Zhu et al., (Cancer Sci. 2013, PMID: 23578198) showed PLK1 dependent resistance to sunitinib depended on inhibition of p53 mediated senescence in ccRCC. Moreover, study by Li et al., (Mol Cancer Ther, 2017, PMID: 28069876) showed PLK1 involved in olaparib (PARP inhibitor) resistance in p53 mutant castration resistance prostate cancer cells. Olaparib treatment clearly caused accumulation of cells at G2/M thus elevation of PLK1, which eventually counteracted the efficacy of Olaparib.

We thank the reviewer for this summary and analyze of our manuscript.

Specific points to be addressed:

1. ccRCC (R10) and ccRCC sunitinib resistant cells (R10R) were treated with BI6727 (100 nM) for cell viability, apoptosis, and H&E experiments (Figure 4 G to I). Results about the apoptosis, cell viability and mitotic phenotype in BI6727 treated ccRCC agree with PLK1 inhibition data published before with other cancer cells line. PLK1 inhibitor at that dose may not just inhibit PLK1 but also other kinases. Generally, it is common practice to apply pharmacological as well as genetic approaches to study loss of function of protein. So, I recommend to use another approach to inhibit PLK1 via i.e siRNA or shRNA in their experiments (Cell viability, apoptosis assay and H&E staining).

We thank the reviewer for this relevant comment. We evaluated the effect of siRNA against Plk1. Like volasertib, Plk1 siRNA decreased cell viability (Fig. 4A) and increased cell death (Fig. 4B) in 498, 786 and 786R cells. Moreover, Plk1 siRNA induced G2M phase accumulation at 48h which is followed by a strong induction of cell death at 96h (SubG1, Fig. 4C).

Figure 4. A and B 498, 786 and 786R cells were transfected with siRNA against Plk1 (siPlk1) for 96h. Cell viability was evaluated by XTT assay (A) and cell death was determined by PI⁺ cells (B). C 786 were transfected with siRNA against Plk1 for 48 h or 96h. The cell cycle was analyzed by flow cytometry.

2. They did not address how PLK1 reversed chemoresistance to sunitinib. I recommend to do senescence Associated (SA) β -galactosidase assay to show how PLK1 bypassed senescence as mechanism of chemo sensitization.

We thank the reviewer for this very interesting suggestion. We performed this experiment. Sunitinib induced senescence in 786 EV but not in 786 Plk1-1 and 786 Plk1-2 cells (Fig. 5). Plk1, by inhibiting sunitinib-inducing p53 activation, probably blocked sunitinib-induced senescence. The results are now included in the revised manuscript in Figure 4F.

Figure 5. 786 EV and 786 Plk1-1 cells were treated with 2.5 μ M sunitinib for 48 h. Senescence was evaluated by β -Galactosidase staining.

3. They showed PLK1 was upregulated in 786R (resistant ccRCC cells) compared to 786 cells (ccRCC cell) (Figure 4F). I recommend to show the cell cycle profile of 786 and 786R (resistance cells) by flow cytometry. Also, I recommend mentioning about p53 status (mutant or WT) in those cells' lines (786 and 786R). Other study showed PLK1 involvement in Olaparib resistance in p53-mutant castration resistance prostate cancer cells. Olaparib treatment clearly causes accumulation of cells at G2/M thus elevation of Plk1, which eventually counteract the efficacy of olaparib.

We thank the reviewer for this relevant comment. 786 and 786R are p53 WT. p53 is rarely mutated in ccRCC. We evaluated the cell cycle of 786 and 786R. No significant difference in cell cycle was observed between 786 and 786R (Fig. 6).

Figure 6. Cell cycle of 786 and 786 overexpressing VHL (786+VHL) were analyzed by flow cytometry.

Validity of any statistical analysis:

It is understandable that in some of their statistical analysis (i.e metastatic ccRCC patients-Belgium cohort Figure 6 B-D) patients number was low (39 to 58 patients). I do not have a big concern on it but would make the result more valid if they have more patients in their analysis (it could be possible that they did not have more patients).

We have increased the number of patients in the Belgium cohort (**from 58 to 158 patients**). The analyzes were performed again and confirmed the results obtained on the 58 initial patients. Nevertheless, and as stated by the reviewer, the statistical analyzes are now much more robust. These results are now included in **Figure 6 in the revised manuscript**.

REVIEWERS' COMMENTS:

Reviewer #1 (Remarks to the Author):

The revised manuscript has been improved with the addition of data addressing the concerns of cell cycle status and hypoxia-dependent effects. The data supports the overall conclusions and observations.

Awkward phrases remain. One example, page 15, line 4.

"This study more specifically addresses the role of Plk1 in ccRCC but we believe that these results can be generalized to several tumors."

Should be role of Plk1 in ccRCC. Also, PLK1 expression in many other tumor types has been well known for more than 10 years.

Reviewer #2 (Remarks to the Author):

The authors have sufficiently addressed all concerns raised by the reviewers of the manuscript including generating extensive new and additional data to do so. It is now acceptable for publication without further revision.